

# Integration and calibration of NDIR CO2 low-cost sensors, and their operation in a sensor network covering Switzerland

Michael Mueller[1], Peter Graf[1], Jonas Meyer[2], Anastasia Pentina[3], Dominik Brunner[1], Fernando Perez-Cruz[3], Christoph Hüglin[1], and Lukas Emmenegger[1]

[1]Empa, Swiss Federal Institute for Materials Science and Technology, Duebendorf, Switzerland
[2]Decentlab GmbH, Duebendorf, Switzerland
[3]Swiss Data Science Center, Zurich, Switzerland

*Correspondence*: Michael Mueller (michael.mueller@empa.ch)

**Abstract.**

More than 300 LP8 $CO_2$ sensors were integrated into sensor units and evaluated for the purpose of long-term operation in the Carbosense $CO_2$ sensor network in Switzerland. Prior to deployment, all sensors were calibrated in a pressure and climate chamber, and in ambient conditions co-located with a reference instrument. To investigate their long-term performance and to test different data processing strategies, 18 sensors were deployed at five locations equipped with a reference instrument after calibration. Their accuracy during 19 to 25 months deployment was between 8 to 12 ppm. This level of accuracy requires careful sensor calibration prior to deployment, continuous monitoring of the sensors, efficient data filtering, and a procedure to correct drifts and jumps in the sensor signal during operation. High relative humidity (> ~85%) impairs the LP8 measurements, and corresponding data filtering results in a significant loss during humid conditions. The LP8 sensors are not suitable for the detection of small regional gradients and long-term trends. However, with careful data processing, the sensors are able to resolve $CO_2$ changes and differences with a magnitude larger than about 20 ppm. Thereby, the sensor can resolve the site-specific $CO_2$ signal at most locations in Switzerland. A low power network (LPN) using LoRaWAN allowed reliable data transmission with low energy consumption, and proved to be a key element of the Carbosense low-cost sensor network.

## 1 Introduction

The number of available low-cost sensor types for ambient trace gas observations has increased in recent years. Frequently, these sensors are combined with wireless data transfer capabilities to form a versatile measurement unit. Low-cost sensors for trace gas measurements are based on different working principles such as metal-oxide semiconductors, electrochemical cells or non-dispersive infrared detection (NDIR). For $CO_2$, NDIR is the most common technique (Lewis, et al., 2018). Similarly to other instruments, knowledge of the sensors' characteristics such as sensitivity, cross-sensitivity or aging is important for meaningful applications. Moreover, the raw sensor output must be converted into molar fraction of the target gas using a mathematical function. The mathematical models provided by the manufacturers are often not sufficient to meet the accuracy demands of trace gas measurements in outdoor conditions. Different approaches such as e.g. multilinear regression (Mueller, et al., 2017; Martin, et al., 2017; Spinelle, et al., 2017), random forest models (Bigi, et al., 2018; Zimmerman, et al., 2018), or artificial neural networks (Spinelle, et al., 2017) are investigated to derive better performing sensor models. However, thorough model validation that is adequate with respect to the foreseen application is necessary for this task, especially as many data-driven models include parameters that were not shown to have a reproducible impact on the sensor signal. Some approaches also employ information in the model that is only valid in a statistical manner, such as for example similar pollutant concentrations at the sensor location and at the closest reference site during selected time periods (Mueller, et al., 2017; Kim, et al., 2018). The use of a standardized terminology for processing levels, as was recently proposed by Schneider et al., (2019), is recommended to clearly define the type of information a sensor model is based on.


The design of low-cost sensors usually relies on a less stable and less controlled measurement environment than high-end instruments. Therefore, the mathematical description of sensor behaviour must be flexible and robust enough to accommodate a wide range of operating conditions. Nevertheless, the accuracy level achieved by low-cost trace gas sensors is still significantly below that of high precision instruments. This may be acceptable in view of their lower costs if the achievable

data quality remains suitable for a specific application. Usually, low-cost sensors have to be individually calibrated for achieving their best performance, and data processing is an essential element to obtain accurate measurements. This data processing includes filtering to eliminate and report outliers or data of reduced quality, and the detection of changes in sensor characteristics which require the adaptation of the model that converts the raw sensor output to molar fraction.

Smart and dependable sensor integration is crucial for both the data quality, and the reliable and cost efficient operation. A

long-lasting autonomous sensor deployment requires that the sensor unit has a low energy consumption, which depends on the energy consumption of the sensing device, the measurement frequency, the on-site data processing, and the method that is used for data transmission. The latter can be achieved using the LoRaWAN protocol (LoRa-Alliance, 2019), which offers data transmission with highly reduced energy consumption compared to mobile communication networks such as GSM, UMTS, LTE.

Increasing the spatial coverage of a measurement network or reducing its costs by the operation of low-cost sensors is appealing. However, the number of long-term applications of low-cost sensors is still sparse (Mueller, et al., 2017; Shusterman, et al., 2016; Castell, et al., 2017; Popoola, et al., 2018). The total costs for the sensors, their calibration, deployment, data transmission, and data processing have to be in equilibrium with the information the sensors provide. Further technical and operational progress is required to enhance this constellation and to integrate more low-cost sensors into meaningful services.

Examples of research activities in the field of lower-cost $CO_2$ measurements and sensor networks are provided by Arzoumanian (2019) and Shusterman (2016).

In this study we present the deployment of more than 250 low-cost $CO_2$ sensors in Switzerland in the framework of the Carbosense project, which aims at assessing anthropogenic and natural $CO_2$ fluxes in Switzerland through the combination of dense observations and high-resolution atmospheric transport modeling. The entire $CO_2$ sensor network is formed by high-

precision instruments, intermediate precision instruments and low-cost sensors. The accuracy of the low-cost sensors is clearly outside the extended compatibility goal of 0.2 ppm for $CO_2$ proposed within the activities of the World Meteorological Organization (WMO) Global Atmospheric Watch (Tans & Zellweger, 2014). However, these sensors are not intended to resolve small regional gradients and trends in atmospheric $CO_2$, They should rather complement the high-precision measurements by providing information on short-term and local variations in $CO_2$ in the order of several tens of ppm as

expected near emission sources, e.g. in the city of Zurich, or due to $CO_2$ accumulation when the boundary layer is shallow.

This paper focuses on the calibration of the LP8 $CO_2$ sensors, their operation within the Carbosense network, the sensor data processing, and the achieved data quality. Most of the findings and developments carried out by means of the Carbosense sensor network such as aspects of data transmission and data processing are generic and transferable to other low-cost trace gas sensor networks.

## 2   Hardware and infrastructure

### 2.1   Carbosense network

The Carbosense $CO_2$ sensor network covers the whole of Switzerland with a regional focus on the city of Zurich (Figure 1). It is formed by three classes of sensors: (i) seven high-precision laser spectrometers (Picarro G1301/G2302/G2401, CRDS), (ii) 20 temperature stabilized, mains powered NDIR low-cost instruments with active sampling and reference gas supply (SenseAir

HPP (Hummelgard, et al., 2015)) and (iii) 300 nodes of battery-powered $CO_2$ low-cost diffusive NDIR sensors (SenseAir LP8). The deployment of the first low-cost sensors started in July 2017, and the network has been continuously extended to





230 sensors by September 2019. The $CO_2$ low-cost sensors are deployed at antenna locations of the telecommunication company Swisscom (4-150 m above ground), at meteorological measurement sites of the Federal Office of Meteorology and Climatology (MeteoSwiss) (10 m above ground), and at sites of the national air pollution monitoring network NABEL (5 m above ground). Within the city of Zurich, the sensors are also mounted on lamp or electricity poles (3-5 m above ground). The

low-cost sensor network covers a wide altitude range from 200 to 2390 m a.s.l., various orographic conditions and landscape types (urban areas, agricultural lands, forests, mountain areas). This implies a wide range of environmental conditions during the operation of the sensors. The deployment of the HPPs started in August 2018 and instruments operate at 15 locations as of September 1, 2019.

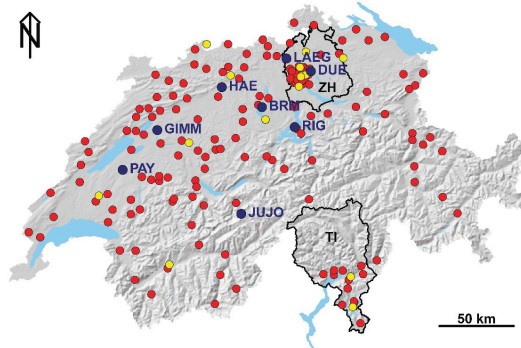

**Figure 1: Carbosense sensor network as of September 13, 2019. Red dots depict LP8 sensor locations, yellow dots depict HPP sensor locations and blue dots depict locations of Picarro instruments. The cantons or administrative divisions of Zurich and Ticino are plotted and marked by ZH and TI. Geographic data used for creating the base map originates from http://www.diva-gis.org and https://www.swisstopo.admin.ch.**

### 2.2    $CO_2$ low-cost sensor unit

#### 2.2.1    Integrated sensors

The $CO_2$ low-cost sensor units (dimensions: 110/80/65 mm) were engineered by Decentlab GmbH (Figure 2). A sensor unit comprises a SenseAir LP8 sensor (SenseAir, 2019), a Sensirion SHT21 sensor (Sensirion, 2019), a LoRaWAN communication

module, a microprocessor, and two batteries for power supply. There is no active ventilation. The LP8 and SHT21 sensors are located close to the opening of the box to ensure fast response times. Dead volumes are kept as small as possible for the same reason. The LP8 sensor reports the infrared measurement (IR), a $CO_2$ molar fraction based on factory calibration, temperature, and its status. The SHT21 sensor measures temperature and relative humidity ($\pm0.3°C$, $\pm2\%$ RH). The measurement frequency was set to 1 minute for all the sensors and the measurements are transmitted as 10 minute averages together with the last single

measurement of the infrared and temperature values over Swisscom's Low Power Network (LPN; based on LoRaWAN). However, during the first weeks of using the sensor units in spring 2017, only the last single values were transmitted for all the measurement types. Since the unit is not equipped with a pressure sensor, pressure has to be measured independently or has to be estimated from other information sources, which is possible with a small uncertainty of $\pm1$ hPa as described in Section 3.2.



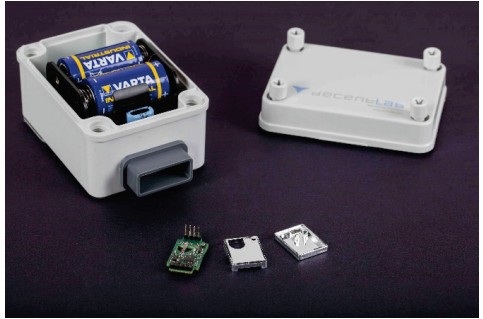

**Figure 2: CO₂ low-cost sensor unit and LP8 sensor (front).**

### 2.2.2 LP8 sensor

Operating conditions of the LP8 sensor are specified by the manufacturer as 0-50°C, 0-85% RH and 0-2000 ppm $CO_2$. The specifications in terms of accuracy are ±50 ppm / ±3% of reading (SenseAir, 2019), which is insufficient for applications in ambient air. The LP8 sensor provides a $CO_2$ measurement based on the factory calibration, the sensor temperature and sensor status information. In addition, the LP8 infrared measurement (preprocessed by the sensor firmware) is accessible. It enables a calibration based on an extended mathematical sensor model that relates the infrared measurement to the $CO_2$ mole fraction $\chi_{CO2}$ in moist air. The parameters of the sensor model have to be determined during a calibration process.

The LP8 is a non-dispersive infrared sensor and, thus, its working principle is based on the Beer-Lambert law.

$$ log\left(\frac{I_0}{I_1}\right) = \epsilon_\lambda \cdot c \cdot d \tag{1} $$

$I_0$ and $I_1$ denote the emitted and detected light, $c$ is the number density of the gas (in units of mol m⁻³), $\epsilon_\lambda$ is the molar attenuation coefficient (m² mol⁻¹), and $d$ is the path length (m) of the beam of light through the cell.

The number of moles of $CO_2$ ($n_{CO2}$) equals

$$ n_{CO_2} = \chi_{CO_2} \cdot \frac{p \cdot V}{R \cdot T} = \chi_{CO_2} \cdot n_{P_0,T_0} \cdot \frac{p \cdot T_0}{p_0 \cdot T} \tag{2} $$

with $p$, $T$ and $V$ denoting the pressure, temperature and volume of the gas, $p_0$ = 1013.25 hPa and $T_0$ = 273.15 K the standard pressure and temperature and $R$ the universal gas constant (8.3145 J K⁻¹ mol⁻¹). With the $CO_2$ number density $c_{CO2} = n_{CO2}/V$ (mol m⁻³) and combining Eq. (1) and Eq. (2) yields

$$ \chi_{CO_2} \cdot \frac{p \cdot T_0}{p_0 \cdot T} = \frac{V}{n_{P_0,T_0} \cdot \epsilon_\lambda \cdot d} \cdot \left(log(I_0) - log(I_1)\right) \tag{3} $$

The volume $V$, the path length $d$ and the molar attenuation coefficient $\epsilon_\lambda$ are unknown constants. Also the emitted light $I_0$ cannot directly be observed. It is expected to slightly change over time. In order to compensate for temperature effects (e.g. through effects on the optical filter or the detector), pressure effects (e.g. through pressure dependent spectral line broadening), and changes in the intensity of the emitted light ($I_0$) or in the geometry of the light beam, Eq. (3) is expanded by additional terms as follows:

$$ \chi_{CO_2} \cdot \frac{p \cdot T_0}{p_0 \cdot T} = k_0 + k_1 \cdot log(I_1) + \sum_{i=1}^{3} u_i \cdot T^i + \sum_{i=1}^{3} v_i \cdot \frac{T^i}{I_1} + w_1 \cdot \left(\frac{p - p_0}{p_0}\right) + f(t) \tag{4} $$

These terms are empirically chosen with the objective to keep the model simple. The coefficients $k_i$, $u_i$, $v_i$ and $w_i$ are unknown and have to be determined by calibration. Temperature effects are described by a polynomial of up to third order, pressure





effects by a linear model. The terms associated with the parameters $v_i$ are based on the transformation $log(I_1+e) = log(I_1 \cdot (1+e/I_1)) = log(I_1) + log(1+e/I_1) \approx log(I_1) + e/I_1$ where $e$ is a small impacting effect.

The function f(t) accounts for possible temporal changes in light intensity $I_0$ or changes in optical path length. For practical

reasons, it was modeled as a step function with a temporal resolution of approximately 14 days during calibration. The variable $T$ is the temperature provided by the LP8 sensor. The $CO_2$ dry air mole fraction $CO_{2,dry}$ can be computed as $CO_{2,wet} / (1- \chi_{H2O})$ with $\chi_{H2O}$ being the air mole fraction of water. This quantity is computed from $T$, $RH$ (SHT21 sensor), and $p$. The used formula is given in the supplement.

For each sensor, the coefficients of Eq. (4) are determined during initial calibration. The final calibrated model describes the $CO_2$ concentration based on $I_1$, $T$ and $p$ accounting for the ideal gas law and additional optical and thermal effects of the sensor. Some of the terms compensating for optical and thermal effects include $I_1$. If $I_1$ changes strongly, the respective compensations are not adequate anymore. Therefore, an additional simplified model is defined with only one term depending on $I_1$. This model has a reduced capability to account for different environmental conditions but it is more robust against large changes in $I_1$.

$$\chi_{CO_2} \cdot \frac{p \cdot T_0}{p_0 \cdot T} = k_0 + k_1 \cdot log(I_1) + \sum_{i=1}^{3} u_i \cdot T^i + w_1 \cdot \left(\frac{p - p_0}{p_0}\right) + f(t) \tag{5}$$

For each sensor and calibration, the coefficients of Eq. (5) are also determined.

The presented LP8 sensor model corresponds to level-2B in the terminology presented by Schneider et al., (2019). This means that, related to the sensor unit, internal and external information is employed but is limited to parameters that are appropriate for artifact correction and directly related to the measurement principle.

### 2.2.3    Data transmission over LPN

The measurements of the sensor units are transmitted every 10 minutes over Swisscom's low power network (LPN) to a central database hosted by Decentlab GmbH. Swisscom's LPN is based on the LoRaWAN protocol (LoRa-Alliance, 2019), using chirp spread spectrum modulation in the frequency band between 863 and 870 MHz, and operating as a commercial service.

LoRaWAN is a wireless network protocol focusing on an asymmetrically organized, energy efficient data transmission. Data can be transmitted as far as several tens of km in rural areas.

In our case, the sensor units have a transmission rate of 10 minutes while the LP8 and SHT sensors operate at a sampling rate of 1 minute. Every transmitted message contains 33 bytes (14 numbers). The energy consumption of data transmission over LPN depends on the spreading factor (SF). Most sensor units in the Carbosense network operate on SF7. In this case, a sensor

unit can independently operate for 5.1 years before the two batteries (alkaline, 1.5V, IEC: LR14) need to be replaced. Here, radio transmission requires 22% of the total energy used by the sensor unit.

### 2.3    Sensor calibration infrastructure

### 2.3.1    Climate and pressure chambers

Calibration data for the determination of the temperature and pressure dependencies of the LP8 sensors were obtained by placing the sensors in climate and pressure chambers. One climate chamber and one pressure chamber at Empa and one pressure chamber at METAS (Federal Institute for Metrology) were used for this task.

In the climate chamber at Empa, the sensors were exposed to at least four 24 hours lasting temperature profiles uniformly decreasing from 50 to -5 °C at $CO_2$ levels of 350, 450, 700 and 1000 ppm. In the pressure chamber at METAS, pressure levels





were varied between 780 and 1050 hPa at $CO_2$ levels of 420 and 900 ppm and at a temperature of 24 °C. In the pressure chamber at Empa, pressure levels were varied between 800 hPa and ambient pressure (approximately 960 hPa) at $CO_2$ levels between 350 and 1000 ppm. The three chambers were not completely air tight, which required a continuous supply of air with a specific $CO_2$ molar fraction, ventilation to ensure a uniform mixture of air within the chambers, and a pump for the pressure

chamber. Picarro G1301/2401 instruments were connected to the chambers for providing $CO_2$ reference values. Pressure was recorded by calibrated instruments (outside the climate chamber, inside the pressure chambers).

### 2.3.2    High-precision CO₂ measurement sites

High-precision $CO_2$ field measurements are performed at several locations in Switzerland. Those used in this project for sensor calibration, assessment of the sensors' long-term performance as well as for correcting the sensor drifts (see section 3.5) are

listed in Table 1 and shown in Figure 1. The $CO_2$ measurement facilities at sites BRM, GIMM and LAEG were initiated within the CarboCount project (Oney, et al., 2015; Berhanu, et al., 2016). Sites HAE, PAY and RIG belong to the Swiss National Air Pollution Monitoring Network, NABEL (Empa, 2018). The $CO_2$ measurement infrastructure at DUE was specifically set up within the Carbosense project to provide an accurate reference for LP8 sensors during ambient calibration. The $CO_2$ measurement instruments were calibrated using working standards with traceability to the WMO-X2007 calibration scale

(Zhao & Tans, 2006; Tans, et al., 2017).

**Table 1: High-precision CO₂ measurements available for this study. The locations of the sites are shown in Figure 1.**

| Site name | Code | Latitude [°] | Longitude [°] | Altitude [m] | Manufacturer | Type | Remark |
|---|---|---|---|---|---|---|---|
| Beromuenster | BRM | 47.18959 | 8.17547 | 798 | Picarro | G2401 | Rural |
| Duebendorf | DUE | 47.40297 | 8.61347 | 432 | Picarro | G2401 | Suburban |
| Gimmiz | GIMM | 47.05345 | 7.24793 | 443 | Picarro | G2301 | Rural |
| Haerkingen | HAE | 47.31187 | 7.82050 | 430 | LI-COR | LI-7000 | Rural, next to a motorway |
| Laegern | LAEG | 47.48196 | 8.39725 | 855 | Picarro | G2401 | Rural, hilltop |
| Payerne | PAY | 46.81308 | 6.94448 | 489 | Picarro | G2302 | Rural |
| Rigi-Seebodenalp | RIG | 47.06739 | 8.46333 | 1030 | Picarro | G2302 | Rural, hillside |

### 2.4    Date storage infrastructure

The raw data from the sensor units, after being transmitted via LPN to a Swisscom server, are forwarded via Internet to Decentlab where they are stored in an Influx database (InfluxDB, 2019) providing near real-time access to the data. Decentlab provides web based dashboards for data visualization as well as APIs for data access in various scripting languages. Information about the sensor network such as deployment history, calibration runs, calibration parameters, observations from reference instruments, and processed sensor measurements is stored in a MySQL database hosted by Empa.

## 3    Data processing

### 3.1    Important issues for LP8 long-term measurements

The deployment of a large number of LP8 sensors in this study revealed two issues that are important for ambient long-term measurements with this sensor type. First, the response characteristics of the LP8 infrared measurement can change over time, both steadily or abruptly. Second, the infrared measurements are susceptible to humidity exceeding a value of about 85%. This

behaviour is common to all LP8 sensors but actual thresholds differ among individual sensors. Therefore, additional processing





steps subsequent to the application of the calibration function are required to achieve a data set of sufficiently high accuracy and completeness.

Several analyses that are presented in the following sections refer to the term deployment. We define deployment as the time period within which a specific sensor unit is placed at a particular outdoor location. A sensor unit can be used in several
consecutive deployments.

### 3.2    LP8 sensor calibration and application of the sensor model

Each LP8 sensor was individually calibrated. For this purpose, each sensor unit was placed in the climate and pressure chambers for at least one complete calibration. Furthermore, each unit was operated under ambient conditions at site DUE until it was shipped for deployment in the Carbosense network. The sensors were run at DUE under ambient conditions in
parallel with a Picarro instrument for a time period between several weeks and several months.

Thus, an extensive data set of both chamber and ambient measurements was collected for each sensor unit to determine the calibration parameters of Eqs. (4) and (5). Filters that exclude conditions near condensation, large changes in IR measurements or in ambient $CO_2$ were applied to this data set for optimal parameter estimation. The data filtering during calibration is more rigorous than the outlier detection applied to the sensors deployed in the Carbosense network (see chapter 3.4). A robust
estimator (Huber loss function) was used for the parameter estimation to minimize the impact of large residuals (e.g. persons breathing near the sensors). The parameters of the LP8 sensor models are stored in the MySQL database. The sensor unit has to pass a new calibration cycle whenever the LP8 sensor is exchanged.

Measurements from LP8 sensors deployed within the Carbosense network are processed by using Eq. (4) and Eq. (5) with the corresponding coefficients determined during the calibration phase yielding a first guess $CO_2$ molar fraction $CO_{2,CAL}$. Thereby,
there are two parallel processing chains but besides the computation of $CO_{2,CAL}$ the further processing is performed equally (outlier detection, drift correction). The function f(t) in Eqs. (4) and (5) is replaced by a constant that equals the last value of this step function during calibration.

The first guess $CO_{2,CAL}$ is subsequently corrected for sensor drifts as described in Section 3.5. Eqs. (4) and (5) require the pressure at the sensor location. This value is derived from 10 minute pressure measurements from the meteorological
measurement network SwissMetNet operated by MeteoSwiss (Supplement Figure 2). A procedure was set up that estimates the vertical pressure gradient in Switzerland and horizontally interpolates the pressure reduced to sea level every 10 minutes. These values allow the computation of the pressure for any location and height above ground level with an uncertainty of about 1 hPa.

Results and flags of subsequent processing steps are stored in the MySQL database to guarantee full traceability and to support
the comparison of different processing options.

### 3.3    Flagging for high relative humidity

A relative humidity threshold $RH_{trsh}$ was determined for every LP8 sensor based on the measurements from the ambient calibration performed at DUE. The purpose is to review the operation limits specified by the manufacturer and to develop a method for flagging the sensor measurements that may be impacted by humidity.
First, the standard deviation of the $CO_2$ residuals (difference between computed $CO_2$ values of the sensors and $CO_{2,moist}$ measured by the Picarro) is computed in 2% RH intervals in a range of relatively dry conditions between 40 and 70% RH (resulting in 15 values in total), and the median of these values denoted as $\sigma_{res}$ is determined. Second, the 95% quantile of the residuals is computed in 2% intervals from 0 to 100% RH. $RH_{trsh}$ is then selected as the maximum interval for which the 95% quantile is smaller than $3 \cdot \sigma_{res}$. The $CO_2$ residuals and the computed $RH_{trsh}$ values are exemplarily depicted for two sensors in
Figure 3 (a) and (b). The operation limits for humidity indicated by the manufacturer (0-85% RH) concur with our results (Figure 3 (c)). All the $RH_{trsh}$ values are stored in the database.

Flagging the measurements of the deployed sensors by applying the criteria RH > $RH_{trsh}$ results in a data set with very few outliers but, concurrently, a significant number of measurements is falsely rejected. In section 3.4 a more adaptive outlier detection algorithm is presented that does not rely on any reference measurements. The choice of the filtering approach depends on the intended use of the measurements and whether the number of undetected outliers or the number of falsely flagged

outliers is more important.

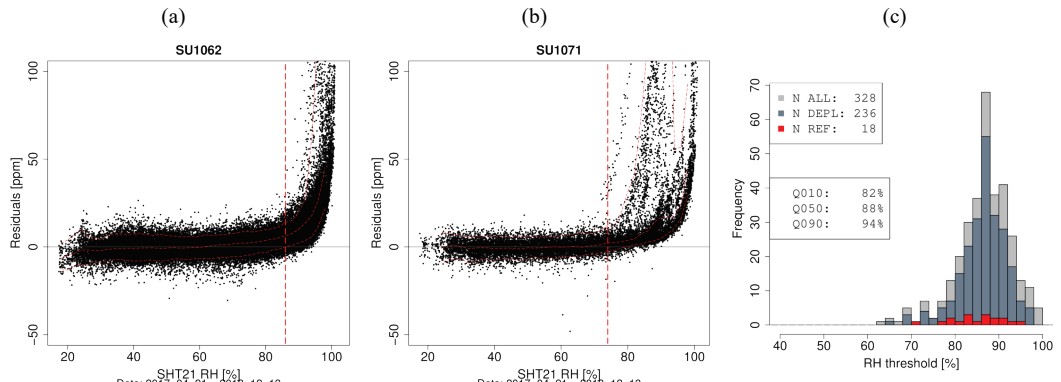

**Figure 3: (a) and (b) $CO_2$ residuals (sensor minus reference; based on Eq. (4)) versus relative humidity during calibration at ambient conditions at the DUE site for sensor units 1062 and 1071. The vertical dashed line indicates $RH_{trsh}$, the other three lines depict the 5%/50%/95% quantiles of the residuals in 2% RH intervals. (c) Overlaid histograms of $RH_{trsh}$ for all the sensors (ALL), within the**
**Carbosense network deployed sensors (DEPL) and at reference sites deployed sensors (REF). The indicated quantiles refer to the set of deployed sensors.**

### 3.4    Outlier detection

We call a LP8 measurement an outlier when it cannot be related to the ambient $CO_2$ molar fraction by means of the sensor
models described by Eqs. (4) and (5). Outliers are primarily caused by relative humidity exceeding about 85% (see Figure 3). Under these conditions the light absorption within the measurement cell can be increased due to the presence of water droplets or condensation of water on the mirrors. Such conditions may last for periods between a few minutes and more than a day. The difficulty in detecting such events is that the signals in the LP8 IR and SHT21 RH time series do not follow a characteristic profile but exhibit significant variation depending on the actual progression of the meteorological conditions. The distinction
between small humidity effects and a true increase in $CO_2$ is not a simple task as the sensor measurements do not fully describe the conditions in the measurement cell. In addition, temporary enhancements from closely located emission sources can unusually impact the $CO_2$ measurements as well and should not be treated as outliers.

The outlier detection algorithm was designed to rely entirely on the measurements from the sensor unit itself and to require no auxiliary information such as measurements from a reference instrument. It analyses and processes quantities derived from
sensor observations that, under normal conditions, vary only slowly. The algorithm learns the sensor's usual behavior at its current location from data obtained during the particular deployment and flags unusual measurements. Prerequisites for the algorithm are that environmental conditions and their changes remain within certain limits and that stable relations exist between specific sensor quantities and environmental conditions. Learning sensor behavior in the field is an important element for minimizing the required calibration time.
Thus, the LP8 outlier detection algorithm is primarily based on the differences of consecutive log(IR) and temperature values plus statistical measures that are derived from a large number of IR measurements. The algorithm also reviews the relative humidity to enhance the robustness of the algorithm. The absolute values of IR and the corresponding values of $CO_{2,cal}$ were





not directly used as both are not stable over time due to drift or jumps and as they depend on $CO_2$, temperature and pressure that are variable over time.

First, the outlier detection algorithm requires the computation of several auxiliary quantities. Here, IR, T and RH denote the infrared measurement, the LP8 temperature and the SHT21 relative humidity. The subscripts M and L refer to the mean and

the last single measurement in a 10 minute interval. $\Delta t$ indicates the time between subsequent measurements transmitted to the database (subsequent measurements are only considered if the difference does not exceed 20 min).

1.  Difference in $log(IR)$:                                                      $\Delta_{IR,M}(t) = log(IR_M(t)) - log(IR_M(t\text{-}\Delta t))$

2.  Difference in $T$:                                                            $\Delta_{T,M}(t) = T(t) - T(t\text{-}\Delta t)$

3.  Mean RH of two measurements:                                                 $M_{RH}(t) = \tfrac{1}{2} \cdot (RH(t) + RH(t\text{-}\Delta t))$

4.  Difference between single measurement and mean:             $\gamma(t) = log(IR_L(t)) - log(IR_M(t)))$

5.  Variance of $log(IR_M(t))$:                                                   $\sigma_M^2(t) = \frac{1}{10} \cdot \frac{1}{n} \cdot \sum \gamma(\tau)^2 \quad \tau \in [t - 2h \dots t - 10min]$

6.  Noise in $log(IR_L)$:

$$IR_{noise} = MAD\big(log(IR_L(\tau)) - log(IR_M(\tau))\big) \quad \tau \in T_{deployment}$$

7.  Median absolute deviation of the difference of consecutive $log(IR_M)$:

$$\Delta IR_{large} = MAD\big(log(IR_M(\tau)) - log(IR_M(\tau - \Delta t))\big) \quad \tau \in T_{deployment}$$

In item 5, n is the number of used $\gamma(t)$ values and 10 is the number of single sensor measurements within a 10 minute interval.

Second, based on the samples in relatively dry conditions (RH<80%), a quadratic function $\Delta_{IR,M} = f(\Delta_{T,M})$ is robustly determined which describes the normal change in $log(IR)$ with a change in temperature. The corresponding residuals $r$ for all samples are

computed and, again based only on the dry samples, the median absolute deviation (MAD) is calculated (Figure 4).

A measurement $IR(t_i)$ is flagged when $|r(t_i)| > 3 \cdot MAD \cap |\Delta_{IR,M}(t_i)| > 3 \cdot \sigma_M(t_i) \cap RH(t_i) > 70\%$ (value set according to Figure 3 (c)). The positive flagged residuals are denoted as $r_{flag,pos}$, the negative flagged residuals as $r_{flag,neg}$. Starting from $t_i$ consecutive (for all the $r_{flag,pos}(t_i)$) or preceding (for all the $r_{flag,neg}(t_i)$) measurements are also flagged until RH drops below $M_{RH}(t_i)$. In general, high relative humidity leads to decreased IR and, concurrently, increased $CO_2$ values for the LP8 sensor. Concurrently,

the sign of r($t_i$) determines the direction of backward or forward flagging of temporally adjacent measurements.

In addition, two more quantities are determined based on $r_{flag,pos}$: $RH_{Q75}$ is the 75% quantile of the RH values and $TDP_{Q25}$ is the 25% quantile of the difference between T and the dew point ($T_d$).

Measurements are also flagged if (i) $|\gamma(t_i)| > 5 \cdot IR_{noise} \cap RH(t_i) > 85\%$, (ii) $|\Delta_{IR,M}(t)| > 5 \cdot \Delta IR_{large}$, (iii) RH > $RH_{Q75}$, or (iv) T $-T_d$ < $TDP_{Q25}$. Under (i), the first criterion is already fulfilled if two of seven $|\gamma(t_i)|$ adjacent to $t_i$ indicate increased noise.


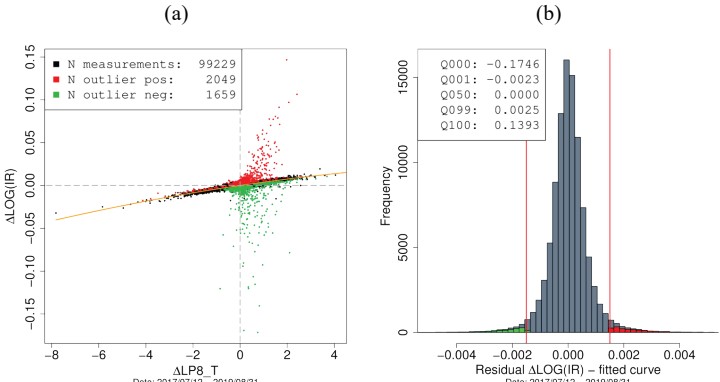

**Figure 4: (a) Differences of consecutive log(IR) values versus differences of LP8 temperatures. Positive outliers are coloured in red, negative outliers are coloured in green. The orange line depicts the quadratic fit of ΔLOG(IR) ~ ΔLP8_T. (b) Histogram of the**





residuals of ΔLOG(IR) with relation to the fitted curve. The vertical red lines depict ±3·MAD. Positive outliers are in red, negative outliers are in green. Results from sensor unit 1010 deployed in Leibstadt are depicted.

### 3.5 Drift correction

IR measurements from LP8 sensors and the corresponding calibrated molar fractions $CO_{2,cal}$ are not stable in time. For sensors deployed in the field, this drift has to be corrected in order to compute unbiased $CO_2$ molar fractions. Since usually no reference measurement is available at the location of the LP8 sensor to determine the drift, a method was developed making use of specific weather conditions during which horizontal gradients in $CO_2$ are small and linking the measurements of the LP8 sensor to those of the closest accurate instrument. The criterion of small horizontal gradients and a well-mixed planetary boundary

layer is best met during situations of high wind speeds.

The drift correction algorithm involves two consecutive steps: First, the identification of time periods $P_{slow}$ when the sensor behavior is slowly evolving, and the drift can be corrected, and of periods $P_{fast}$ when the behaviour changes abruptly. Second, the determination of the drift and its correction.

For the first task, the identification of $P_{slow}$, the calibrated measurements $CO_{2,CAL}$ from the afternoon are analyzed because $CO_2$

molar fractions are most comparable from day to day in the afternoon when the planetary boundary layer is usually well mixed (Supplement Figure 1).

The algorithm computes for each sensor and day $t_d$ the following quantities from the calibrated measurements $CO_{2,CAL}$:

1.  $Q_{prev7d}(t_d)$:   20% quantile of $CO_{2,CAL}(\tau)$ where $\tau \in$ [t-7d…t-1d] ∩ $\tau \in$ [13:00-17:00 UTC]
2.  $Q_{next7d}(t_d)$:   20% quantile of $CO_{2,CAL}(\tau)$ where $\tau \in$ [t+1d…t+7d] ∩ $\tau \in$ [13:00-17:00 UTC]

3.  $Q_{prev15d}(t_d)$:   20% quantile of $CO_{2,CAL}(\tau)$ where $\tau \in$ [t-15d…t-1d] ∩ $\tau \in$ [13:00-17:00 UTC]
4.  $Q_{next15d}(t_d)$:   20% quantile of $CO_{2,CAL}(\tau)$ where $\tau \in$ [t+1d…t+15d] ∩ $\tau \in$ [13:00-17:00 UTC]
5.  $Q_{15d}(t_d)$:   20% quantile of $CO_{2,CAL}(\tau)$ where $\tau \in$ [t-7d…t+7d] ∩ $\tau \in$ [13:00-17:00 UTC]
6.  $b_{15d}(t_d)$:   slope of $CO_{2,CAL}(\tau)$ where $\tau \in$ [t-7d…t+7d] ∩ $\tau \in$ [13:00-17:00 UTC]

(Time in Switzerland refers to CET/CEST.)

Further, an empiric threshold $\Delta Q_{TRSH}$ is computed as median($Q_{n7d} - Q_{p7d}$) + 5·MAD($Q_{n7d} - Q_{p7d}$). Sensor behavior is considered unsteady ($P_{fast}$) if $|Q_{next7d}(t_d) - Q_{prev7d}(t_d)| > \Delta Q_{TRSH}$ or if $|b_{15d}(t_d)| > 3$ ppm/day ∩ $|Q_{next15d}(t_d) - Q_{prev15d}(t_d)| > 40$ ppm. Drift correction is applied separately to the measurements of each sensor deployment and continuous time period $P_{slow}$. An example of the working principle of the algorithm is shown in Figure 5.





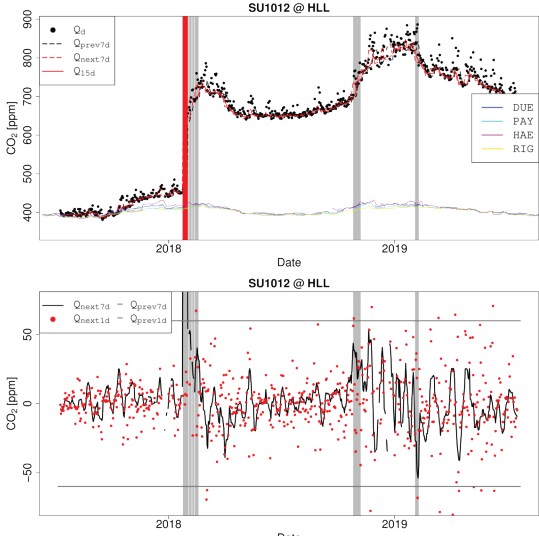

**Figure 5: (a) Time series of $Q_d$, $Q_{prev7d}$, $Q_{next7d}$ and $Q_{15d}$ for sensor unit 1012 deployed in Hallau (HLL). The red vertical lines depict days when $|Q_{next7d}-Q_{prev7d}|$ is larger than the threshold. Shaded periods indicate time periods with increased $|b_{15d}|$. For comparison, the $Q_{15d}$ values for the reference sites DUE, PAY, RIG and HAE are shown. (b) Time series of $Q_{next7d}-Q_{prev7d}$ and $Q_{next1d}-Q_{prev1d}$ for the same sensor. The horizontal lines depict the threshold $\Delta Q_{TRSH}$.**

The actual drift correction is based on wind measurements from MeteoSwiss sites (Supplement Figure 2) and $CO_2$ measurements from the high-precision instruments deployed in the network (both 10 minute averages). The drift correction algorithm is applied to the calibrated measurements $CO_{2,CAL}$ from the sensors deployed in the Carbosense network.

First, all the MeteoSwiss sites within a distance of 40 km from a sensor are selected. Time periods are identified when all the selected sites report for at least 90 minutes

  i.  wind speed > 2 m/s, or

  ii. wind speed > 0.75 m/s $\cap$ median(wind speed at selected sites) > 3 m/s

Time periods lasting longer than 4 hours are split into shorter intervals with a duration of about 2 hours each. Second, the most closely located $CO_2$ reference is chosen (Figure 1). Its data is checked for completeness (number of measurements $n \geq 6$) and variability (sd $\leq 4$ ppm) within each windy period. Similarly, the sensor data is checked for completeness ($n \geq 6 \cap$ SHT21 RH<$RH_{trsh}$) and variability (sd $\leq 15$ ppm). Third, the $CO_2$ offset $\Delta CO_2(t)$ between the sensor's and the reference's median is computed for each windy period, and a continuous $CO_2$ offset time series is derived by linear interpolation between these periods. Drift corrected sensor measurements are derived by adding the linearly interpolated $\Delta CO_2(t)$ to the measurements $CO_{2,CAL}$.

For the data set presented in this study, we use only the high precision measurements from the sites DUE, PAY and GIMM for the adjustment of the LP8 sensors. This procedure allows quantifying the accuracy of the concept by means of the remaining reference sites. In fact, measurements from GIMM are only used to adjust LP8 sensors deployed in PAY. Thereby, co-located sensor and reference measurements are independent in this data set (see section 4.2). Obviously, three reference sites are not sufficient to accurately adjust all the LP8 sensors deployed in Switzerland as weather conditions often differ from region to region. Drift correction for a final and optimized LP8 data set will rely on measurements from all the reference sites and also from the HPP instruments (Figure 1).



The assumption of spatially homogeneous $CO_2$ mole fractions during strong wind events was tested by treating measurements from reference instruments in the same way as those from the LP8 sensors. Whenever an LP8 sensor would be corrected at sites BRM, LAEG, HAE and RIG relative to DUE or at site PAY relative to GIMM, the 10 minute $CO_2$ molar fractions measured by the Picarro instruments at the two sites are compared. Not considered are RH and measurement completeness of

the LP8 sensor. Figure 6 shows $CO_2$ differences of measurements from sites LAEG (distance d=19 km; height difference $\Delta$h=423 m), RIG (d=39 km; $\Delta$h=598 m), BRM (d=41 km; $\Delta$h=366 m) and HAE (d=61 km; $\Delta$h=-2 m) with respect to DUE as well as $CO_2$ differences of PAY (d=35 km; $\Delta$h = 57 m) with respect to GIMM. All these sites are located in or adjacent to the Swiss plateau (Figure 1 and Figure 6 (f)) and therefore have mostly similar weather conditions. The $CO_2$ differences are depicted in two histograms placed on top of each other. The histograms in light grey shows all 10 minutes $CO_2$ differences,

while the histograms in dark grey only present those differences during windy conditions. The concept works well for background sites (LAEG, RIG, BRM) but has limitations for sites that are locally impacted by emissions (HAE is located next to a motorway). For all the site pairs, the differences of the $CO_2$ measurements show a small bias (-2.1 – 0.8 ppm) and a scatter component (2.2 – 2.8 ppm at background sites, 6.0 ppm at the traffic site HAE). The RMSE of the differences amounts to 2.3 to 3.6 ppm (background site) and 6.2 ppm (traffic site). The situation for HAE can be improved if the effect of local emissions

is reduced and only measurements between 22:00 and 04:00 UTC and/or wind directions upward the motorway are selected (Supplement Figure 5). Obviously, that concurs with a reduction of the number of adjustment periods. An additional time filter to reduce the effect of local traffic emissions is applied to three sites in the Carbosense network that are located next to a busy road.

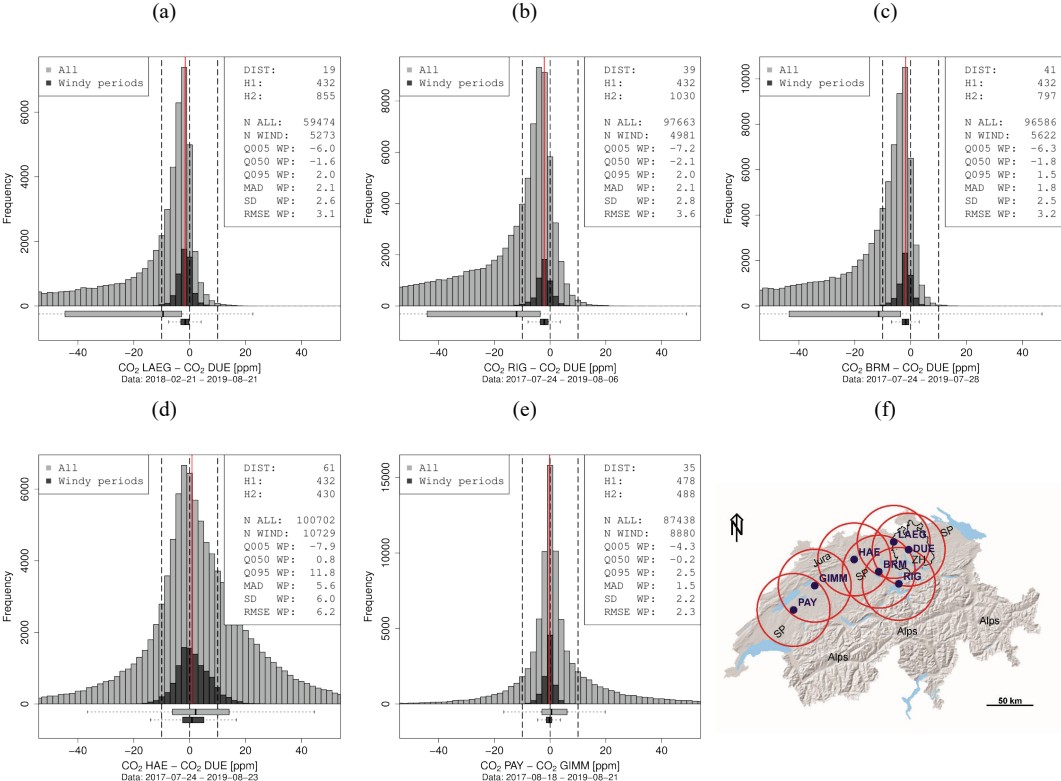

**Figure 6: $CO_2$ differences of measurements at (a) LAEG, (b) RIG, (c) BRM and (d) HAE with respect to DUE and $CO_2$ differences of measurements at (e) PAY with respect to GIMM. DIST denotes the distance between the two sites [km], H1 and H2 denote the altitudes of the two sites [m]. Q005 WP, Q050 WP, Q095 WP denote the 5%, 50% and 95% quantile of the $CO_2$ differences in windy conditions, respectively. MAD WP and SD WP denote the median absolute deviation and standard deviation of the $CO_2$ differences**





in windy conditions. RMSE WP denotes the RMSE of the CO₂ concentrations of the two sites in windy conditions. (f) Map of the locations of the reference sites, their 40 km perimeters and the names of geographic regions. SP: Swiss Plateau, ZH: Canton of Zurich. Geographic data used for creating the base map originates from http://www.diva-gis.org and https://www.swisstopo.admin.ch.

### 3.6 Consistency check

There are instances where the LP8 sensor drift cannot be corrected as frequently as required. This can be caused by extended meteorological situation with low wind speeds or by sensor related issues (e.g., unstable behavior, simultaneous wind and high relative humidity). Consequently, the difference between the computed and the true $CO_2$ molar fraction may increase over

time. In addition, the outlier detection algorithm can be less effective during prolonged time periods with no dry conditions.

In order to identify such periods of suspicious or less accurate data, the measurements of individual sensors were checked for consistency with the more accurate measurements from HPP and Picarro instruments in a similar geographic setting. Although the true $CO_2$ mole fractions at a given site are unknown, $CO_2$ time series of sites within a particular region are expected to exhibit similarities, e.g. similar daily $CO_2$ minima in the afternoon when the boundary layer is usually well mixed.

For this purpose, all the locations of the Carbosense network were divided into three groups based on their region and the surrounding topography.

All the sites in the Canton of Ticino (Figure 1) are part of group one as only two HPPs are operating in this region. The sites in the other regions of Switzerland are divided into two groups depending on whether they are located on a hilltop (group 3) or not (group 2). A hilltop location is defined by the following criteria: (i) The difference in altitude, i.e. the topography in a

2.5 km perimeter including the actual altitude of the mounted sensor, is larger than 300 m. (ii) More than 90% of the topography in a 2.5 km perimeter is at a lower altitude than it is at the sensor location. The second criterion is omitted if the difference in altitude exceeds 400 m.

The $CO_2$ molar fraction of the reference instruments and the HPPs are analyzed group by group. The 10 percent quantile of the preceding 24 hours is computed for each instrument/HPP every sixth hour ($CO_{2,Q10\%}$). Afterward, a band is derived

($CO_{2,limits} = \mathrm{median}(CO_{2,Q10\%}) \pm 2.0 \cdot \mathrm{range}(CO_{2,Q10\%})$) that indicates plausible daily minimum $CO_2$ molar fractions. The preceding 24 hours of measurements from a sensor get flagged in case the sensor's daily $CO_2$ minimum is outside the computed band.

### 4 Results

### 4.1 Sensor calibration

The employed sensor model that is based on the Beer-Lambert law and is extended by an empirical parametrization can relate the sensor IR measurements and the ambient $CO_2$ molar fraction in all relevant $CO_2$, temperature and pressure conditions (Figure 7 (a), (b), (d), (e)). The sensor's factory calibration is intended for using the sensor in a narrower temperature range like it is encountered indoors and does not include pressure information. Measurements based on the factory calibration are not accurate enough for observations under outdoor conditions (Figure 7 (c)). The 1/IR-terms in Eq. (4) require that the sensor's

IR measurement does not heavily drift or jump because in this case the error cannot be compensated by a simple offset. The data quality of sensors whose IR values significantly jumped during deployment (~ 300 ppm in $CO_2$) is usually better when the simplified sensor model (Eq. (5)) without 1/IR-terms is applied (see Figure 11). Eq. (5) provides a less optimal fit under particular operating conditions (e.g., for high $CO_2$ molar fractions) (compare with Figure 7 (a) and (b)). However, this is of minor importance for most locations. The $CO_2$ molar fraction at locations which are not impacted by nearby emissions is

usually within 380-550 ppm. Temperature effects cause by far the largest deviation of the sensor response from the ideal gas law (Figure 7 (c), (d), (f)). Pressure effects are of a much smaller magnitude (~ 0.1 ppm/hPa).

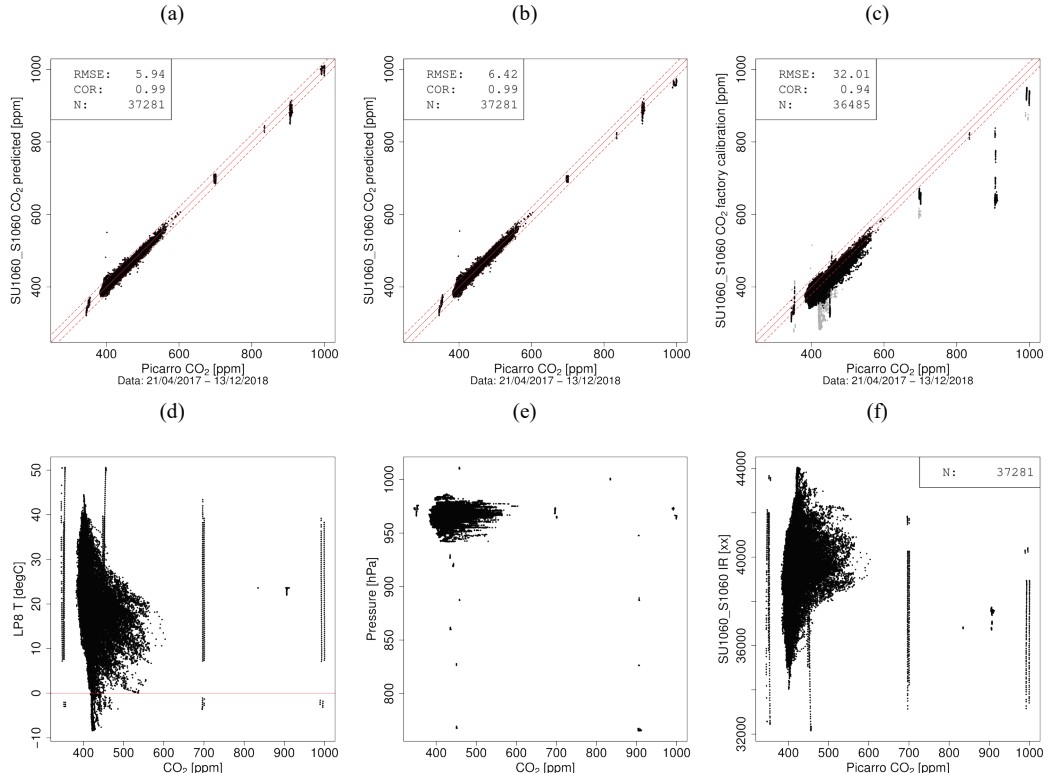

**Figure 7: (a) Calibrated sensor measurements (Eq. 4) versus measurements from Picarro instruments exemplarily shown for sensor 1060. The data set contains measurements from the climate and pressure chambers and ambient measurements. The band between the red dashed lines denotes a range of ±20 ppm. (b) Same as in (a) but for Eq. (5). (c) Same as in (a) but for factory calibrated sensor measurements. Measurements outside the sensor specifications are depicted in gray and included for RMSE/correlation. (d) $CO_2$ – LP8 T plot and (e) $CO_2$ – P plot depicting the environmental conditions covered during calibration. Same data set as in (a). (e) LP8 IR measurements versus $CO_2$ from the Picarro instruments. Same data set as in (a).**

As shown in Figure 8, the RMSE of the LP8 $CO_2$ measurements with respect to the Picarro during chamber and ambient calibration is between 6.8 and 12.5 ppm when applying Eq. (4) and between 8.0 and 13.9 ppm when applying Eq. (5) for the deployed sensors. Data filtering during calibration is chosen to be very selective in order to optimally determine the sensor model parameters.





(a)               (b)

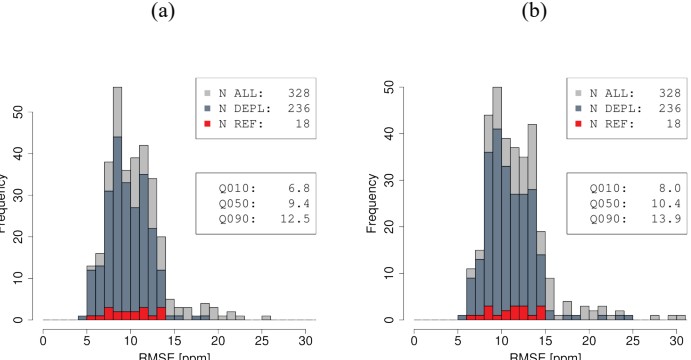

**Figure 8: RMSE values of sensor calibration (a) using Eq. (4) and (b) using Eq. (5). Three histograms are overlaid: all calibrated sensors, sensors deployed in the Carbosense network (DEPL) and sensors at locations with a reference instrument (REF). The indicated quantiles refer to the set of deployed sensors.**

### 4.2 Drift correction and outlier detection

The performance of the outlier detection and drift correction algorithms is presented together as both processing steps have to be applied for obtaining accurate $CO_2$ measurements. The results shown in this section refer to sensor measurements in the period July 1, 2017 to September 1, 2019.

Several sensor units are operated at sites equipped with a $CO_2$ reference instrument (HAE: 5 sensor units; PAY: 5 sensor units; RIG: 5 sensor units; LAEG: 2 sensor units; BRM: one sensor unit) in order to test different calibration and processing options. Drift correction for the sensors in PAY relies on the $CO_2$ measurements from GIMM and for the sensors in RIG, HAE, LAEG and BRM on the $CO_2$ measurements from DUE (Figure 6). Thus, the sensor and reference instrument measurements are independent at these sites.

A slightly modified data processing scheme was applied to the data from the 141 sensor units that were operated in DUE longer than until December 1, 2017. This additional data treatment provides the opportunity to assess the data quality for a larger set of sensors. The calibration data set for these sensors contains all data before December 1, 2017 and is applied to the measurements thereafter. The sensor data are processed as described in chapter 3 but drift is corrected by referring to measurements from sites LAEG and BRM instead of DUE (site LAEG being located closer to DUE is used when both instruments provide data). The accuracy of $CO_2$ molar fraction from these sensors located in DUE can therefore be compared to that from sensors deployed in the Carbosense network. Among the sensors in DUE there are also those with a performance that is not sufficient for deployment and therefore they are held back in DUE.

The comparison of the median difference between $CO_2$ measurements from the sensors and from the reference instruments reveals that sensor drift can be adjusted over the long-term when the sensor measurements can regularly be referred to $CO_2$ predictions (Figure 9). The frequency of the required adjustments depends on the individual sensor as the change in sensor behavior and the corresponding drift are not evolving constantly.

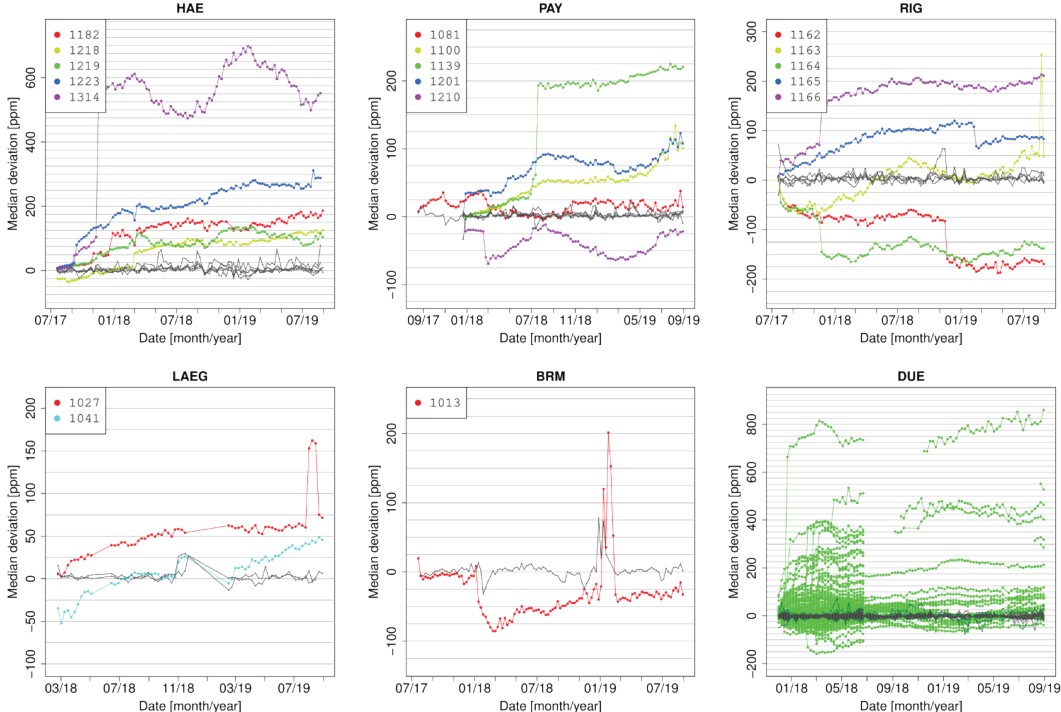

**Figure 9: Weekly median deviation of sensors operated in HAE, PAY, RIG, LAEG, BRM and DUE before (coloured) and after drift correction (grey). Note the different scales in the y-axis.**

By means of the sensors operating co-located with reference instruments the effect of different processing options can be

assessed. This includes the employed sensor model (Eq. (4) or Eq. (5)), the applied outlier detection (no outlier detection, outlier detection based on $RH_{trsh}$ or the algorithm presented in section 3.4) and the use of additional consistency checks. The sensor and reference measurements are compared for weekly periods by means of the root mean square error (RMSE) and the correlation (Figure 10 (a) and (b)). In addition, the fraction of valid measurements w.r.t. the total number of measurements in the database is indicated (Figure 10 (c)). It shows the effect of data filtering on the number of usable measurements. Scatter

plots of the comparisons between the LP8 measurements and the measurements from the reference instruments at HAE, PAY, RIG, LAEG and BRM are shown in Figures 6 to 9 in the supplement.

The long-term accuracy of the sensors amounts to 8-12 ppm on average. The accuracy can be reduced in particular time periods. This value strongly depends on the data filtering. A rigorous data filtering using $RH_{trsh}$ leads to the best RMSE values. The outlier detection algorithm performs slightly worse in terms of the RMSE. Overall, it classifies a slightly larger number

of measurements as valid than the filtering using $RH_{trsh}$. Differences in performance between the sensor models described by Eq. (4) and Eq. (5) are small for this set of sensors. The accuracy of the measurements can be further improved when they are validated against measurements from high-precision instruments operated in the Carbosense network. This is shown for the combination of the outlier detection algorithm and the consistency check. Correlation between sensor and reference is about 0.9 on average. At sites RIG, LAEG and BRM the correlation coefficients are smaller due to smaller $CO_2$ variations

encountered at these locations (Supplement Figure 1).





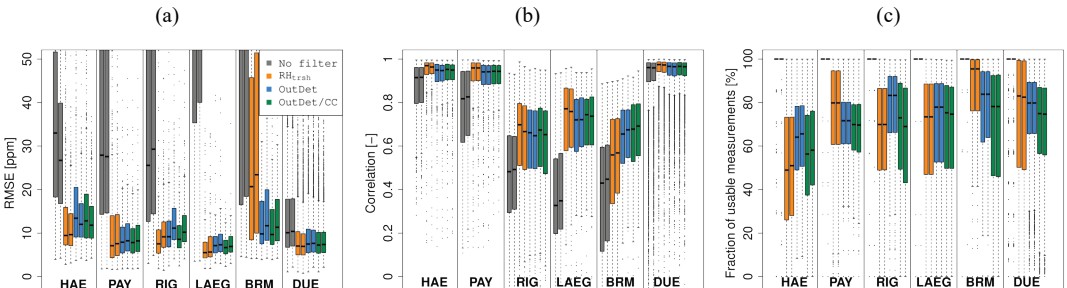

**Figure 10: (a) Weekly RMSE values for all the sensors deployed at HAE, PAY, RIG, BRM/LAEG and DUE. For each site four versions are presented for the drift adjusted measurements: (i) no filtering applied, (ii) outlier detection based on sensor specific $RH_{trsh}$ value, (iii) outlier detection algorithm, and (iv) outlier detection algorithm plus consistency check. The left bar with the same colour refers to Eq. (4), the right bar to Eq. (5). (b) Same as in (a) for the weekly correlation. (c) Same as in (a) for the weekly fraction**
5  **of used data. Here, the fraction refers to the number of measurements transmitted to the database.**

The extended sensor model described in Eq. (4) is applicable for a wider range of environmental conditions ($CO_2$, T, P) than the reduced version (Eq. 5). However, when the IR signal shows large changes (> |300| ppm expressed in molar fraction) as in the case of sensor unit SU 1314 deployed at HAE, the application of the simplified sensor model provides more accurate results
10  (Figure 11).

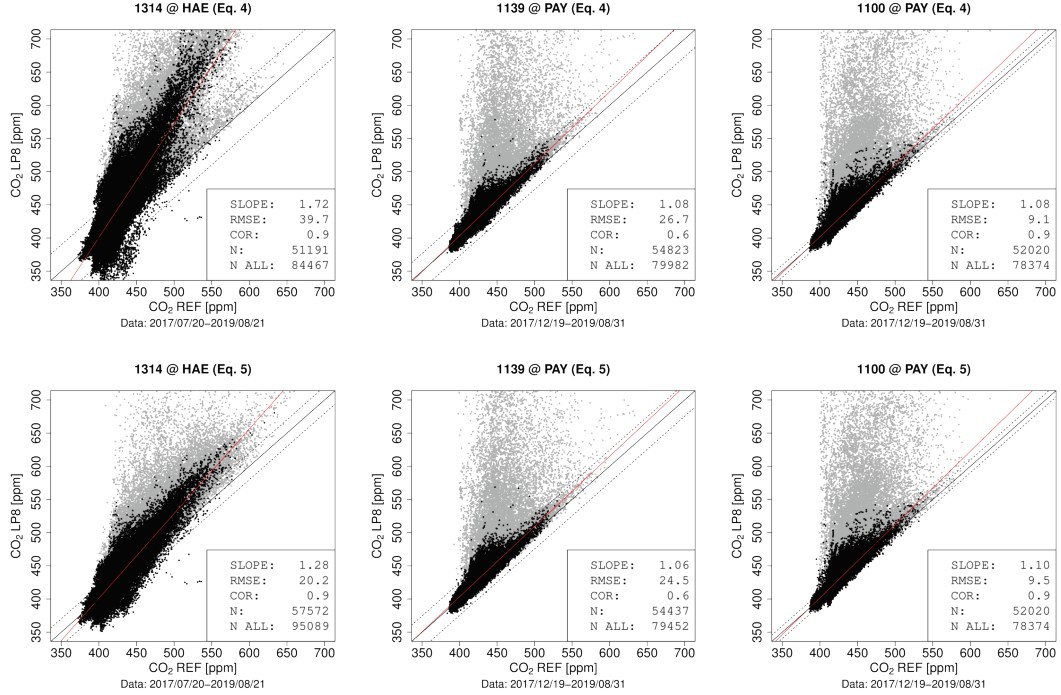

**Figure 11: Comparison of sensor and reference $CO_2$ measurements for SU 1314 deployed in HAE and SU 1100 and SU 1139 deployed in PAY. The sensor measurements depicted in the figures in the top row are based on the sensor model given by Eq. (4) and in the bottom row they are based on the sensor model given by Eq. (5). The sensor measurements are drift corrected, and the outlier detection algorithm was applied. Points in grey are outliers.**



### 4.3 Differences between co-located sensors

Co-located sensor units are an additional option to assess the sensor performance. They reveal how similarly two sensors behave when they encounter comparable environmental conditions. There are 12 locations where two sensor units operate in parallel but where no reference instrument is available (Figure 12 and Figure 6 of the supplement). Horizontal distance between

5  the sensor unit pairs does not exceed 45 m. There are no close emission sources for cases with distance $\neq 0$. The indicated RMSE refers to the difference of simultaneous measurements. The sensor pairs exhibit fairly good correlations at most locations. For the sensor pairs operated in HLL and BSCR there is better agreement when processing the measurements using the sensor model given by Eq. (5) instead of the model given by Eq. (4). The IR measurement of sensor unit 1012 changed significantly in January 2018, those of sensor unit 1120 changed significantly in March 2018. The difference between the

10  processing models is small for the other sensor pairs.

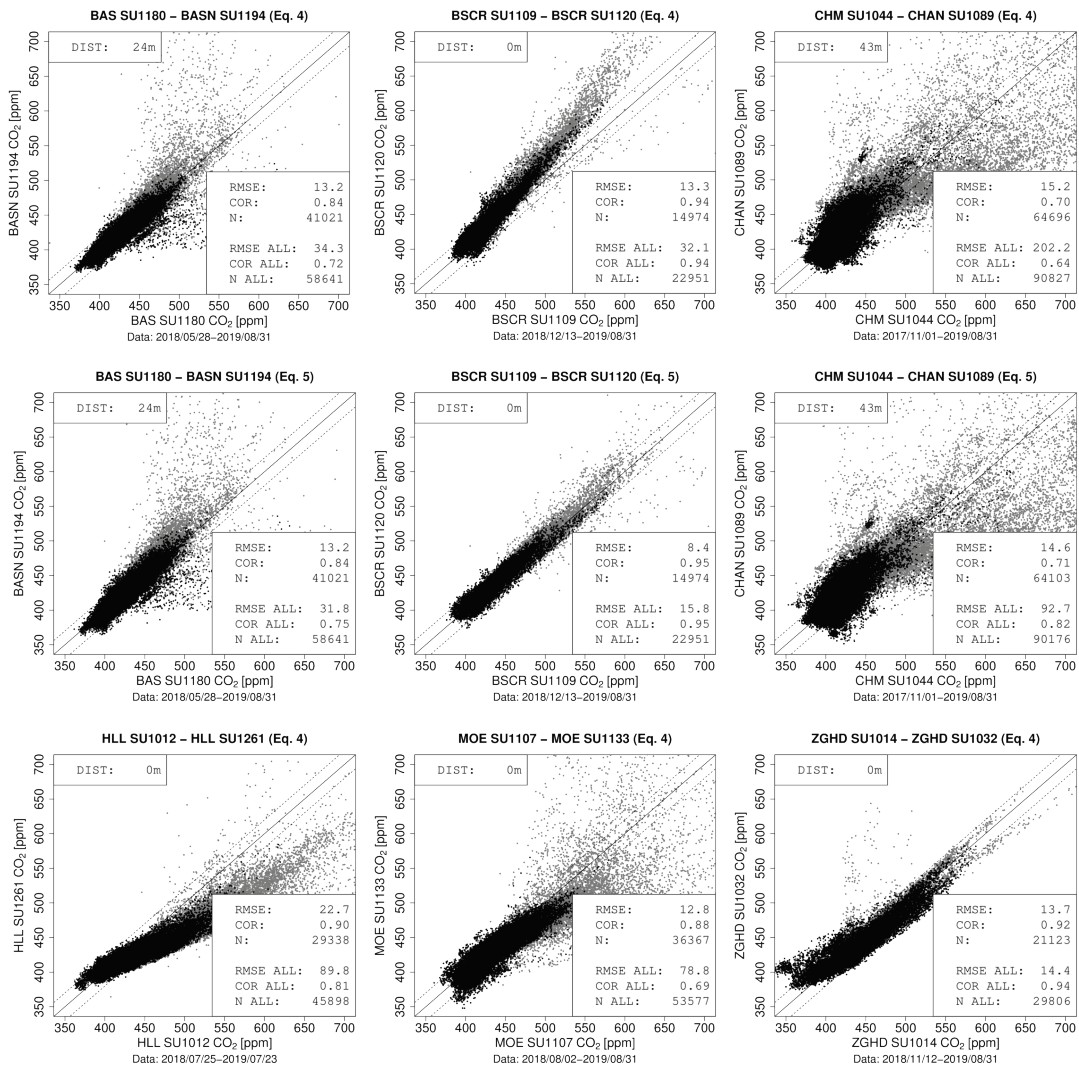



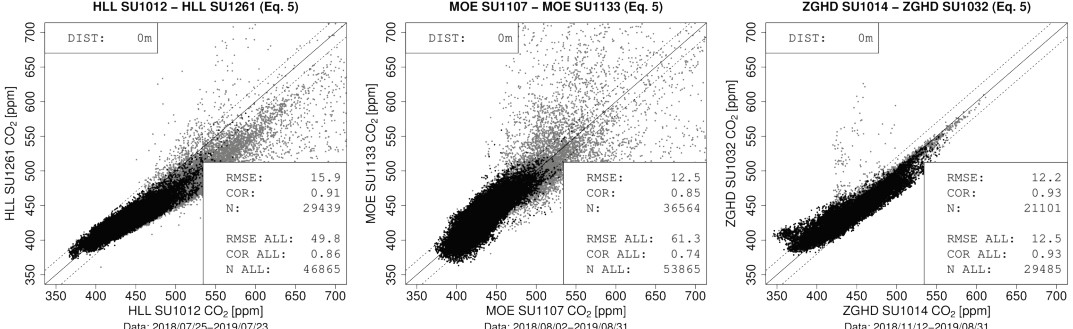

**Figure 12: Comparison of LP8 measurements (drift corrected, outlier detection algorithm) from co-located sensors (distance between sensors < 45 m). Points in grey are flagged as outliers. The header of the individual figures indicates the sensor pairs by the location name and the sensor unit ID as well as the sensor model.**

Eight sensor units were deployed in the Carbosense network and they were brought back to site DUE to review their performance due to sensor malfunctioning (e.g. LP8 sensor dropped out of the board) or suspicious $CO_2$ measurements. For completeness, the comparison between the measurements from these sensors and from the Picarro instrument is shown in the supplement.

**4.4    Overall data coverage**

The Carbosense network consists of 230 LP8 sensors as of 1 September 2019. In total, there were 262 deployments in the period July 1, 2017 to September 1, 2019. Over 75% of the deployments lasted longer than one year and five lasted less than 30 days.

The data transmission over Swisscom's Low Power Network (LPN) works reliably. The 25%/50%/75% quantiles of the
fraction of transmitted data for individual deployments at MeteoSwiss and NABEL locations and at locations within the city of Zurich amount to 90%/95%/98% (Figure 13 (a)). Performance is even better at Swisscom's transmitter locations (25% quantile: 98%). However, these are usually equipped with a LPN gateway and built at elevated locations. We cannot assess to which part of the data transmission process the data loss is attributed (transmission module used in the sensor unit, LPN infrastructure, LPN network coverage). The transmission module (Microchip RN2483) of several sensor units was found to
have a reduced reliability at high temperatures (above about 30º C).

A small number (~1%) of the transmitted LP8 measurements had a nonzero status flag, for instance, when temperature was below -8.5º C (LP8 specific threshold) or the sensor is malfunctioning. For a minor fraction of measurements a drift adjustment could not be performed as the sensor was assessed to be in an unstable phase. The outlier detection algorithm flags 23% of the measurements that were drift corrected. In combination with the consistency check 29% of the measurements are flagged.
There is considerable variability in these fractions related to the individual sensor performance and the location. A clear relation is evident between the fraction of outliers and the humidity conditions encountered at the deployment location (Figure 13 (b)). Overall, the median of usable measurements from all individual deployments amounts to 67%. There is a diurnal variation in the fraction of flagged measurements closely related to the diurnal variation in relative humidity (Figure 14 (a)). The outlier detection algorithm has the advantage of retaining a larger number of measurements in conditions of high relative humidity
compared to the method using $RH_{trsh}$ (Figure 14 (b)).

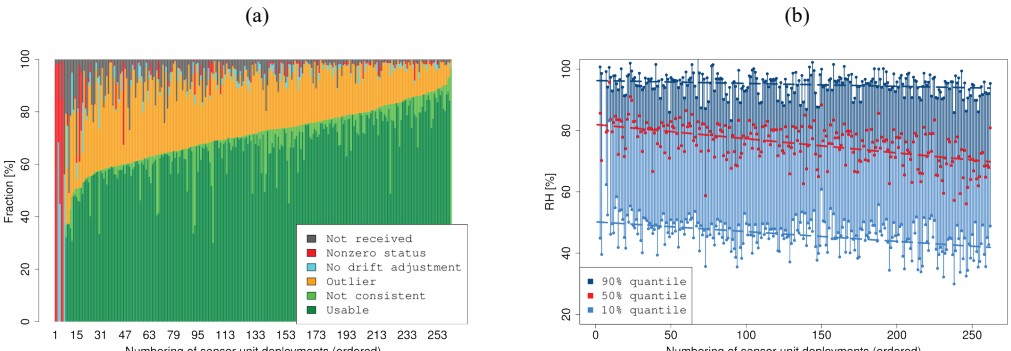

Figure 13: (a) Analysis of measurement yield in the Carbosense network. Gray: Difference between expected and actual number of measurements in the database. Red: Measurements transmitted to the database with nonzero LP8 status (e.g., temperature below - 8.5° C, sensor malfunctioning). Cyan: Measurements with no drift adjustment (e.g. periods with unstable sensor behaviour). Orange: Measurements flagged by outlier detection. Light green: measurements that did not pass the consistency check. Dark green: Usable measurements. (b) Distribution of the measured relative humidity. Ten, fifty and ninety percent quantiles of RH. Ordering equally as in (a).

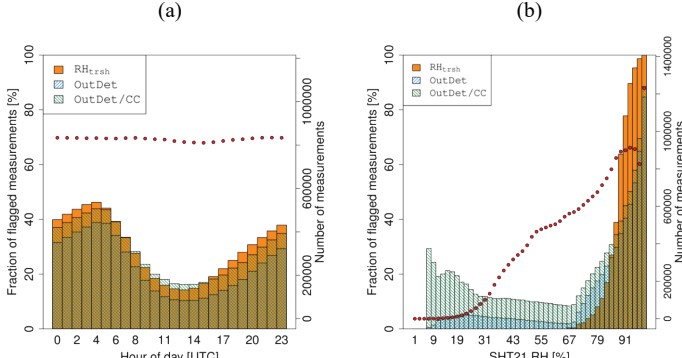

Figure 14: Analysis of the results of measurement filtering referring to time of day (a) and relative humidity (b). Filtering is based on (i) a sensor specific $RH_{trsh}$ value, (ii) the outlier detection algorithm (OutDet), and (iii) the outlier detection algorithm plus a consistency check (OutDet/CC). For the calculation of the fraction of flagged measurements, the number of measurements and flags of all deployments are summed. The numbers of measurements are depicted as red dots.

### 4.5 Computation of the water volume fraction

The conversion of wet $CO_2$ to dry $CO_2$ requires the water molar fraction $\chi_{H2O}$. This value is computed for the sensor units based on the SHT21 T and RH measurements and the pressure that is interpolated for the specific location. The uncertainty in the estimation of $\chi_{H2O}$ and the corresponding uncertainty in the dry air mole fraction of $CO_2$ can be assessed for a total of 55 sensor units operated at MeteoSwiss SwissMetNet sites that are equipped with more accurate meteorological instruments. At those sites, $\chi_{H2O}$ has been computed from the sensor units and from reference T, RH and p measurements (Supplement Figures 3 and 4). The agreement is best (±0.07%) when global radiance is low (<50 W/m²). In this case T and RH measured inside the box are representative for the outside conditions. Deviation is slightly worse (±0.15%) for higher global radiance. For the majority of the measurements, the conversion of wet $CO_2$ to dry $CO_2$ molar fraction is associated with an error below 1.2 ppm (assessment of deviation for an error $\epsilon = 0.2\%$ and $\chi_{CO2,wet} = 600$: $\chi_{CO2,dry} = \chi_{CO2,wet} / (1-\epsilon/100) = 600$ ppm / $(1-0.2/100)) = 601.2$ ppm).





## 5 Discussion and conclusions

Calibration, drift correction and outlier detection are crucial elements for the operation of the LP8 sensors in a sensor network. Due to the number of employed sensors and the slight differences in their individual response characteristics the processing scheme has to be optimized in terms of accuracy, yield of usable measurements and processing efficiency. As the processing scheme consists of several independent elements each of them can be further improved in the future.

The sensor calibration reveals the dependencies of the sensor signal on $CO_2$, temperature and pressure as well as the impact of environmental conditions such as humidity on the measurements. The mathematical sensor model has to account for a varying sensor response over time. Our approach is to use an extended model as long as the sensor behavior does not drift significantly. After large jumps in the IR signal, sensor measurements can be processed based on a simpler sensor model to optimize the measurement accuracy until the sensor is replaced.

We present two methods for the detection of outliers. The application of the two methods for individual sensors leads to a different number of flagged measurements and concurrently to different RMSE values. Flagging the measurements based on a conservative RH threshold results in most accurate results. The presented outlier detection algorithm that relies on no reference measurements is similarly powerful. The possibility to learn individual sensor characteristics in the field is an important feature to reduce calibration time.

The response of the LP8 sensors is not stable over time and frequent adjustments are required. The performed correction during windy periods works well for the regions in and adjacent to the Swiss plateau (Figure 6). The method relies on a dense network of meteorological observations and high-precision $CO_2$ measurements. Moreover, it strongly depends on the prevailing meteorology and, therefore, it is prone to a shortage of suitable adjustment periods. This situation could possibly be enhanced by using the results of an operational atmospheric transport model. Two aspects are expected to be improved by using such a model: (i) the identification of time periods when the $CO_2$ molar fraction in the atmosphere is homogeneous and sensors and reference instruments can be related and (ii) the determination of the vertical $CO_2$ gradient. Such an atmospheric transport model is currently under development at Empa and its applicability for the sensor network will be investigated.

The data processing for sensors in the Swiss Plateau and especially in the region of Zurich (Figure 1) where the Carbosense network is most dense is operational. For these regions, the analysis of measurements from reference sites shows that drift correction within selected time periods works well. Results from atmospheric transport models will be required to achieve a similar data quality for the sensors located in the Swiss Alps.

The LP8 sensor measures the $CO_2$ concentration with an accuracy of 8 – 12 ppm on average if the sensors are calibrated, continuously monitored and drift corrected during operation, and the measurements are filtered. Nevertheless, the accuracy was found to be reduced in limited time periods. The LP8 sensors are well capable to resolve differences in $CO_2$ concentrations exceeding 20 ppm ($2 \cdot \sigma$). $CO_2$ variations encountered at locations in Switzerland usually exceed this threshold (see Figure 1 of the supplement). Exceptions are high altitude locations such as Jungfraujoch (3580 m a.s.l.) (Sturm, et al., 2013). Near-surface $CO_2$ signals depend on anthropogenic emissions, the activity of the biosphere (uptake, respiration) and meteorology (boundary layer height, transport of $CO_2$). LP8 sensors can resolve the site-specific $CO_2$ signals for a wide range of locations, from





elevated background sites to sites next to motorways (Figure 15). The usability of the sensors clearly reaches its limits when small signals or small long-term trends shall be detected.

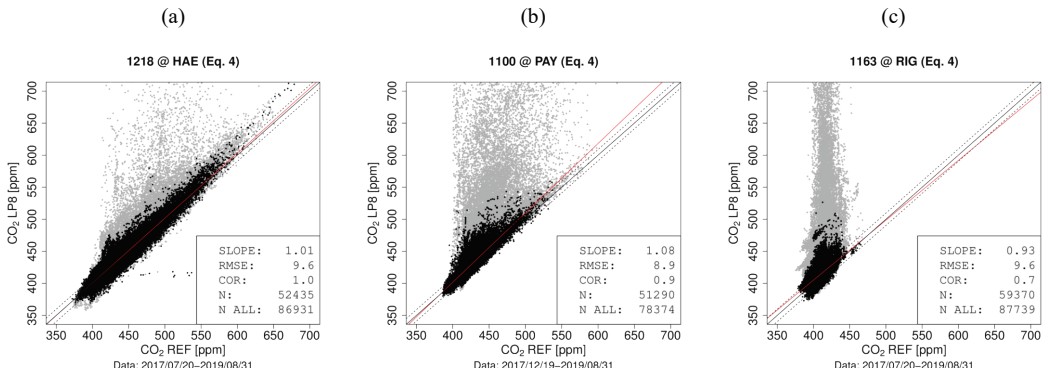

**Figure 15: Comparison between LP8 and reference measurements. The LP8 measurements are outlier screened, drift corrected and checked for consistency. (a) Site HAE is located next to a motorway, (b) site PAY is located in a rural landscape, and (c) site RIG is an elevated background site. Points in grey are outliers.**

**Data availability.**

Periodic data releases on the ICOS Carbon Portal are in preparation. Temperature and RH measurements from the sensor units of the period July 1, 2017 to October 1, 2019 are already available under https://doi.org/10.18160/RW69-MP2Y.

**Competing interests.**

Author Jonas Meyer is the CTO of DecentLab, the manufacturer of the sensor units. Jonas Meyer was solely involved in development and manufacturing of the sensor units and in the transmission of the raw sensor data into the database hosted by DecentLab and was not involved in sensor deployment or in data analysis. The authors declare that they have no other competing interests.

**Acknowledgements.**

We acknowledge MeteoSwiss, Swisscom, the Swiss National Air Pollution Monitoring Network (NABEL), the Environment and Health department of the City of Zurich (UGZ), Agroscope, and the Amrein Futtermühle AG for their generous support of the Carbosense sensor network. In addition, we acknowledge Swisscom's contribution to the sensor units and data transmission. We are grateful for SenseAir's support of the project. We thank Markus Leuenberger (University of Bern) for providing $CO_2$ measurements from the sites Beromuenster and Gimmiz. Günter Grossmann (Empa) is acknowledged for his support in the operation of the climate chamber.

Funding is provided by the Swiss Data Science Center (SDSC) through the project CarboSense4D and the Swiss State Secretariat for Education, Research and Innovation (SERI) through project Eurostars E!11401 CO2.GLOBAL.

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
