# Peer review of "Integration and calibration of NDIR CO2 low-cost sensors, and their operation in a sensor network covering Switzerland"

_Atmospheric Measurement Techniques, 2019_

## Referee Comment (RC1) · Anonymous Referee #1 · 15 Dec 2019

The authors present here an excellent evaluation and use-case demonstration of a low-cost CO2 sensor platform and an observing network comprised of said sensors. I think the work presented here is very thorough and covers all the bases needed to determine the usefulness of data archived by this network.

I have no major issues/concerns or comments but I have a couple of small things that I would like the authors to think about / comment on in a response:

1) You mention how important pressure measurements are, eg. in the description of how the ideal gas law is used to get CO2 mole fractions. While the methodology of using adjusted sea level pressure from a nearby meteorological station and interpolating from the $CO_2$ sensor height above ground level is probably sufficient, I can't help but wonder why air pressure measurements are not made as part of the sensor package? There are multiple small, low-cost pressure sensors that would be by far the most accurate measurements in the package (sub 1 hPa accuracy). Perhaps something to include in the second-generation of the instrumentation package.

2) What are the theories / reason(s) for why there are sudden large jumps in the reported $CO_2$ values from the sensors? Is it degradation in the IR lamp/sensor? I would think generally that would be more gradual than seemingly instantaneous. Or are there other failures that still allow for usable observations, but need that offset correction?

Overall, I think this is an excellent study and look forward to the final published version soon.

---

## Referee Comment (RC2) · Anonymous Referee #2 · 20 Dec 2019

Review of "Integration and calibration of NDIR CO2 low-cost sensors, and their operation in a sensor network covering Switzerland" for AMT, by Mueller et al.

The manuscript describes the calibration, deployment, data processing, and data quality filtering for a network of low-cost sensors deployed in Switzerland. The authors have put a large amount of effort into this process and clearly a lot of careful thought into achieving the most high-quality data product one can get from these sensors, and this is to be applauded. This should be published in AMT, as this information provided here can guide many researchers across the world interested in deploying low-cost CO2 sensors. My main issue is with the final pronouncement of the uncertainty of the

sensor data, which is stated to be 20 ppm at 2-sigma but not explained. Many different statistics are shown in the paper, but it's not clear which one is being used to assess the overall possible accuracy of the sensor network. Otherwise, all my comments are minor and only needed for clarification.

Comments:

L10: LP8 should be defined (a commercial low-cost non-dispersive infrared (NDIR) CO2 sensor)

fig S6-S9: caption or legend should explain gray vs. black points.

P2L9: awkward wording. should read "..is crucial for both high data quality and reliable and cost-efficient operation"".

P2L19: what constellation? perhaps the authors meant "a low-cost sensor network".

P2 L28 comma should be period?

P3, lines 2: Earlier (P2 L39) both the HPP and the LP8 were described as "low-cost", which are you referring to here?

Same comment P3 L18. Perhaps it would be better to not describe the HPP's as low-cost on P2 L39 (medium-cost instead, perhaps), to avoid this confusion, as I believe this refers to the LP8 (next sentence makes that clear).

Fig. 2, it is hard to see where the LP8 resides within the white box. could a second panel be added here with a schematic? I still wonder about response time. It is hard to see what kind of opening there is.

Eq 1-4: chi_co2 is not defined at all here. Presumably mole fraction, from equation 2. should be stated along with units. And this of course is not the dry mole fraction, right? Please state this.

p5 L6: Here CO2,wet is the same as chi_co2 above? keep consistent.

P5 L6, only this dilution effect is needed to account for water vapor? Is there no additional effect of water on the measurement (as there is in CRDS for example)? Perhaps point to Fig. 3 here which shows that there is no RH dependence below the threshold RH.

P5,L11 CO2 mole fraction (not concentration?)?

P5, L17 - which of these two is used finally, eq 4 or 5? [Later this explained, that both are used and evaluated - perhaps note this here to avoid confusion].

Table 1 - is the altitude the altitude of the sensor above sea level (i.e. elevation + height of sampling pole), or the elevation of the site? It would be useful to have both. Perhaps this is addressed later, but were the LP8's colocated on the same sampling line as the high-precision instruments (either next to the sampling inlet or pulling from the same line?).

Section 2.4 title should be "data" not "date". P6 L20 = all sensor units or just the LP8 units?

P6 L30 - Reference to Section 3.3 would be nice here for the reader to know it's coming.

P7L10 - to ask my previous question again, out of curiosity, how was the "in parallel" achieved? The LP8 sensor units do not draw air in through an inlet, so presumably the inlet to the Picarro was located very close to the opening in the LP8 sensor box?

P7, L21, I'm not super clear on this f(t) step function. Perhaps a figure illustrating how it's determined during calibration?

P7 L18, How is this uncertainty of the pressure interpolation known? (has this pressure been evaluated against a measurement somewhere)?

P8L14: should be "an LP8".

Section 3.5 (and maybe elsewhere): please be consistent between CO2,cal and CO2,CAL (case of subscript).

Also, clarify this quantity relative to the equations earlier (page 4 & 5): is this CO2, dry from p5, line 6, which is derived from Co2,wet as computed using equations 4 or 5 (which one?), where in those equations it is referred to as x_co2?

Perhaps remind the reader here that CO2,CAL is the calibrated value using the laboratory calibration from the chamber and co-location experiments where the parameters to those equations (4 &5) were determined.

p11L15, does this include night-time periods, or only 13-17?

p11, L20 Can the authors include a figure of this time series? It would be nice to see if drift is typically long-time scale monotonic drift, or if it is variable in time from one to the next windy period, indicating that the drift may not be captured by this method due to the (in)frequency of windy periods? Or to show if maybe a linear fit in time to this offset might work better?

p12 L15 remind us what this is in local time?

p12 Fig 6 & text, this is a nice analysis to indicate the accuracy of your calibration during windy periods. Although would the authors expect many of the LP8 sites to behave more like HAE, as they are often found in urban areas at low height?

p12 L18 - is this what is meant by this sentence, that some of the LP8 sites are treated a bit differently with additional filtering prior to comparing w/ high-precision sites? Perhaps another sentence would explain this better - time filter on wind direction?

P13 L33 awkward phrasing. Perhaps, "like that encountered indoors, and does not include a pressure correction" (or the effect of pressure in the calibration).

P13 L34: "not accurate enough" is not really quantitative - one could say that 20 ppm is not accurate enough either. Rephrase perhaps, that it is not as accurate as can be achieved with this sensor as you show in (a) and (b)?

Fig 9: indicate that the different colours represent different individual sensors when all

[Figure]

are co-located (this took me a while to realize from the legend).

P16 L12: "long-term accuracy", what quantity exactly is being reported here? The range of the deviations from the reference instrument after drift correction? Accuracy is not supposed to be a quantity, it is qualitative, like "good" or "bad". I think here it should state "error" or even better, explain the exact quantity that is calculated here. (mean difference, range of differences over all the sensors, etc.). Later references to accuracy are valid (e.g. " the accuracy can be improved".. etc. is fine).

p19 L25: relation should read "relationship"

p21 L9: This is misleading - the calibrations themselves did not account for RH dependence; rather the water vapor correction was applied, and the comparison with high-precision data showed the RH dependence was not there until a threshold was reached. Perhaps cite Fig. 3 here.

P21 L34. The authors have done such a great amount of work to evaluate these sensors, that to boil it all down to this range seems to do it diservice. Please state what this number is and how it was calculated.

P21 L37: And where is this 20ppm shown? It is not clear where this comes from at 2-sigma - is it from the RMSE shown in the figures?

P22 L1: last sentence is awkward.

Data Availability: I encourage the authors to ensure the data is full available prior to publication. That is my understanding of the rules of this journal.

SI, page 11 there is a non-English word there!!

---

## Referee Comment (RC3) · Anonymous Referee #3 · 21 Dec 2019

This is a thorough and detailed analysis of the deployment of a low-cost CO2 sensor network supported by a number of reference grade instruments. The paper substantially adds to the literature and advances the conversation about the potential of such networks and the effort required to sustain them.

It will be interesting to see whether all of the NDIR sensors available have similar performance characteristics as the one described here or whether there are some that have more stable in-field characteristics. If the authors have insight or speculation into that question it would be interesting to know their opinion.

---

## Author Comment (AC1) · 19 Feb 2020

**Reply to the reviewers' comments on the manuscript "Integration and calibration of NDIR CO2 low-cost sensors, and their operation in a sensor network covering Switzerland"**

Michael Mueller1, Peter Graf1, Jonas Meyer2, Anastasia Pentina3, Dominik Brunner1, Fernando Perez-Cruz3, Christoph Hüglin1, and Lukas Emmenegger1

1Empa, Swiss Federal Institute for Materials Science and Technology, Duebendorf, Switzerland
 2Decentlab GmbH, Duebendorf, Switzerland
 3Swiss Data Science Center, Zurich, Switzerland

Correspondence: Michael Mueller (michael.mueller@empa.ch)

**10**

We thank the referees for their reports and their valuable contributions to the improvement of the manuscript. For each reviewer, we list all of his or her comments (RC) followed by the authors' replies (AR) and the performed changes in the manuscript. The indications of pages and lines refer to the originally submitted paper version.

**Reviewer #1**

- 15 RC: You mention how important pressure measurements are, e.g. in the description of how the ideal gas law is used to get CO2 mole fractions. While the methodology of using adjusted sea level pressure from a nearby meteorological station and interpolating from the CO2 sensor height above ground level is probably sufficient, I cannot help but wonder why air pressure measurements are not made as part of the sensor package? There are multiple small low-cost pressure sensors that would be by far the most accurate measurements in the package (sub 1 hPa accuracy). Perhaps something to include in the second-20 generation of the instrumentation package.
- 20 generation of the instrumentation package.

**AR:** We agree with the reviewer that the integration of a pressure sensor would improve the sensor units, especially its independency from access to pressure data from a meteorological measurement network or from a numerical pressure modelling system. We can report that the second generation of the  $CO_2$  sensor units manufactured by Decentlab are equipped with a low-cost pressure sensor. The accuracy of the sensor amounting to 0.5 hPa is sufficient for this type of application.

Changes: None.

**RC:** What are the theories / reason(s) for why there are sudden large jumps in the reported  $CO_2$  values from the sensors? Is it degradation in the IR lamp/sensor? I would think generally that would be more gradual than seemingly instantaneous. Or are there other failures that still allow for usable observations, but need that offset correction?

AR: The sensors operate in a wide range of environmental conditions (T, RH) while they are deployed. High diurnal variations in T and RH are frequent. We observe both, gradual changes in the sensor signal (drift) and sudden jumps. The reason for the large jumps is not entirely known but very likely due to mechanical stress of the plastic housing under the above mentioned changes in environmental conditions. This can lead to abrupt changes in the optical path and finally the sensor signal. Changes: We added a sentence in P6L29 that states the supposed cause for the large jumps.

**35 Reviewer #2**

The manuscript describes the calibration, deployment, data processing, and data quality filtering for a network of low-cost sensors deployed in Switzerland. The authors have put a large amount of effort into this process and clearly a lot of careful thought into achieving the most high-quality data product one can get from these sensors, and this is to be applauded. This should be published in AMT, as this information provided here can guide many researchers across the world interested in deploying low-cost CO2 sensors. My main issue is with the final pronouncement of the uncertainty of the sensor data, which is stated to be 20 ppm at 2-sigma but not explained. Many different statistics are shown in the paper, but it's not clear which one is being used to assess the overall possible accuracy of the sensor network. Otherwise, all my comments are minor and only needed for clarification.

**AR:** We agree with the reviewer that the information on how the sensor accuracy is determined should be given in more detail. We address this topic below where the reviewer specifically addresses this subject.

Comments:

**RC:** L10: LP8 should be defined (a commercial low-cost non-dispersive infrared (NDIR) CO2 sensor)

**Changes:** We rephrased the first sentence of the abstract.

RC: Fig S6-S9: caption or legend should explain grey vs. black points.

**Changes:** We rephrased the captions of Figures 6 to 9 of the supplementary materials and explain the difference between black and grey points.

**RC:** P2L9: awkward wording. should read "..is crucial for both high data quality and reliable and cost-efficient operation"". **Changes:** We changed this sentence as suggested by the reviewer.

**RC:** P2L19: what constellation? perhaps the authors meant "a low-cost sensor network".

Changes: We rephrased the sentence to improve clarity.

RC: P2L28 comma should be period?

Changes: We replaced the comma by a period.

- 65 RC: P3, lines 2: Earlier (P2 L39) both the HPP and the LP8 were described as "low-cost", which are you referring to here? Same comment P3 L18. Perhaps it would be better to not describe the HPP's as low-cost on P2 L39 (medium-cost instead, perhaps), to avoid this confusion, as I believe this refers to the LP8 (next sentence makes that clear).
  Changes: We rephrased the sentence starting at P2L37, i.e. we changed "low-cost instruments" by "medium-cost sensors".
- 70 RC: Fig. 2, it is hard to see where the LP8 resides within the white box. Could a second panel be added here with a schematic? I still wonder about response time. It is hard to see what kind of opening there is. Changes: We added a schematic to Figure 2 that shows the location of the LP8 sensor within the sensor unit.

**RC:** Eq 1-4: chi\_co2 is not defined at all here. Presumably, mole fraction from Eq. 2 should be stated along with units. And this of course is not the dry mole fraction, right? Please state this.

AR: The parameter chi\_co2 is defined in line 10 of page 4 of the submitted manuscript. However, this information is also important for the following paragraph, so we repeat it. The parameter chi\_co2 has units [mol]/[mol].Changes: We repeat the definition of chi\_co2 after Eq. 2 along with the information that it refers to moist air.

RC: p5 L6: Here CO2, wet is the same as chi\_co2 above? keep consistent.AR: That is true.

Changes: We changed CO2, dry to chi\_co2, dry and CO2, wet to chi\_co2 in order to keep consistent.

RC: P5 L6, only this dilution effect is needed to account for water vapour? Is there no additional effect of water on the
measurement (as there is in CRDS for example)? Perhaps point to Fig. 3 here which shows that there is no RH dependence
below the threshold RH.

**AR:** The LP8 sensor measures  $CO_2$  in moist air as the air is not dried before the measurement. The sentence on P5L6 describes the transformation of  $CO_2$  molar fraction in moist air to  $CO_2$  molar fraction in dry air. Sensor measurements (SHT21 T/RH) and the interpolated pressure are required for this computation. The indicated transformation is important as atmospheric models usually use molar fraction in dry air as input.

We cannot completely rule out an impact of water vapour on the LP8 measurements at RH values below 85 %. However, some testing of parametrizations relying on RH in the sensor model equations did not lead to clear improvements. **Changes:** We added a sentence before sentence P5L6 that mentions that atmospheric models usually rely on CO2 in dry air.

**RC:** P5,L11 CO2 mole fraction (not concentration?)?

**RC:** P5L17 - which of these two is used finally, Eq. 4 or 5? [Later, it is explained that both are used and evaluated - perhaps note this here to avoid confusion].

**Changes:** We rephrased the sentence at P5L16. We state now that for each sensor and calibration the coefficients of Eqs. (4) and (5) are determined.

**RC:** Table 1 - is the altitude the altitude of the sensor above sea level (i.e. elevation + height of sampling pole), or the elevation of the site? It would be useful to have both. Perhaps this is addressed later, but were the LP8's co-located on the same sampling line as the high-precision instruments (either next to the sampling inlet or pulling from the same line?).

- AR: The LP8s were deployed in close horizontal distance (1-3 m) to the inlets of the tubes that provide air to the high-precision instruments. At HAE, PAY and RIG, the inlet is on the roof of the air quality monitoring station. At DUE, the inlet of the tube that connects to the Picarro instrument is next to the rack where the LP8s are calibrated at a height of about 2 m. Site LAEG is a tall tower. Here, the LP8s and the inlet of the tube to the Picarro instrument are both located on 28 m above ground. At
- 110 site BRM, the LP8s are located on the roof of the air quality monitoring station (5 m) and the lowest inlet of the tube that connects to the Picarro instrument is at 12.5 m. Note that six of these seven sites are rural sites without emission sources in close distance and only one site is a traffic site. Therefore,  $CO_2$  molar fraction should be nearly identical within horizontal distances of less than 10 m.

**Changes:** We added the information about the heights of the inlets of the tubes that connect to the high-precision instruments and the heights of the LP8 sensors, if any are present at the particular station, in table 1. The caption of the table was adapted.

RC: Section 2.4 title should be "data" not "date". Changes: We changed "date" to "data".

- RC: P6 L20 = all sensor units or just the LP8units?
   AR: All the measurements from all the LP8 sensor units and from all the HPP sensor units are transmitted over LPN. The measurements from the high-precision instruments are not transmitted over LPN.
   Changes: None.
- RC: P6 L30 Reference to Section 3.3 would be nice here for the reader to know it's coming.Changes: We added a reference at the end of the sentence on lines P6L30 to P7L2 that points to sections 3.3, 3.4 and 3.5.

**RC:** P7L10 - to ask my previous question again, out of curiosity, how was the "in parallel" achieved? The LP8 sensor units do not draw air in through an inlet, so presumably the inlet to the Picarro was located very close to the opening in the LP8 sensor box?

**AR:** See reply related to table 1.

Changes: See changes made in table 1.

RC: P7, L21, I'm not super clear on this f(t) step function. Perhaps a figure illustrating how it's determined during calibration?
AR: The step function f(t) is a set of temporally consecutive constants that are only non-zero in specific time periods [t=t1..t2]. Changes: None.

**RC:** P7 L18, How is this uncertainty of the pressure interpolation known? (has this pressure been evaluated against a measurement somewhere)?

AR: We computed the accuracy of the interpolation method with a leave-one-out cross-validation based on the measurements from the MeteoSwiss SwissMetNet sites. Moreover, we compared the interpolated pressure to pressure measurements from the HPP units supporting the results of the LOO-validation.

Changes: We added this information in a sentence in line P7L28.

**RC:** P8L14: should be "an LP8".

Changes: We changed "a" to "an".

**RC:** Section 3.5 (and maybe elsewhere): please be consistent between CO2,cal and CO2,CAL (case of subscript). **Changes:** Now, the parameter is throughout labelled as CO2,CAL.

**RC:** Also, clarify this quantity relative to the equations earlier (page 4 & 5): is this CO2, dry from p5, line 6, which is derived from Co2, wet as computed using equations 4 or 5(which one?), where in those equations it is referred to as  $x_{co2}$ ? Perhaps remind the reader here that CO2, CAL is the calibrated value using the laboratory calibration from the chamber and co-location experiments where the parameters to those equations (4 & 5) were determined.

**Changes:** We added the information about the correspondence between CO2,CAL and chi\_co2 from Eqs. 4 and 5 in the sentence in line P10L5.

RC: p11L15, does this include night-time periods, or only 13-17?AR: The detection of windy periods are not limited to the afternoon.

Changes: None.

**RC:** p11, L20 Can the authors include a figure of this time series? It would be nice to see if drift is typically long-time scale monotonic drift, or if it is variable in time from one to the next windy period, indicating that the drift may not be captured by this method due to the (in)frequency of windy periods? Or to show if maybe a linear fit in time to this offset might work better?

better?

**AR:** For many sensors the drift is mainly monotonic for most of the time. Though, several sensors experience sudden jumps. Therefore, the monitoring of and the accounting for the sensor's drift behaviour is necessary. Possibly, the combination of "measurements" (CO2 offset estimated during a windy period) and "model" (monotonic drift) in a Kalman filter framework could provide slightly improved results. Such approaches will be explored in the near future.

**Changes:** We included an additional Figure in the manuscript that shows the  $\Delta CO_2$  time series for the sensors deployed in the canton of Zurich. We restricted the presented time series to that group of sensors as comparability between the sensors and the clarity of the Figure is optimal.

**RC:** p12 L15 remind us what this is in local time?

Changes: We added in the sentence in line P12L15 the information that time in Switzerland refers to CET/CEST.

**RC:** p12 Fig 6 & text, this is a nice analysis to indicate the accuracy of your calibration during windy periods. Although would the authors expect many of the LP8 sites to behave more like HAE, as they are often found in urban areas at low height? p12 L18 - is this what is meant by this sentence, that some of the LP8 sites are treated a bit differently with additional filtering prior to comparing w/ high-precision sites? Perhaps another sentence would explain this better - time filter on wind direction?

- **AR:** The assumption of similar  $CO_2$  molar fraction at two locations during windy periods breaks down when closely located emissions sources significantly contribute to the measured concentrations. Only a few of the sensors located in the city of Zurich are located at busy roads. As mentioned in P12L6, sensor corrections at these sites are only performed during windy periods in nighttime.
- 185 Changes: We reformulated the sentence in line P12L16 to describe the subject clearer.

**RC:** P13 L33 awkward phrasing. Perhaps, "like that encountered indoors, and does not include a pressure correction" (or the effect of pressure in the calibration).

Changes: We rephrased this sentence based on the reviewer's suggestion.

**RC:** P13 L34: "not accurate enough" is not really quantitative - one could say that 20 ppm is not accurate enough either. Rephrase perhaps, that it is not as accurate as can be achieved with this sensor as you show in (a) and (b)? **Changes:** We rephrased this sentence based on the reviewer's suggestion. 195 RC: Fig 9: indicate that the different colours represent different individual sensors when all are co-located (this took me a while to realize from the legend).

**Changes:** We changed the caption of Fig. 9 as suggested by the reviewer.

RC: P16 L12: "long-term accuracy", what quantity exactly is being reported here? The range of the deviations from the 200 reference instrument after drift correction? Accuracy is not supposed to be a quantity, it is qualitative, like "good" or "bad". I think here it should state "error" or even better, explain the exact quantity that is calculated here.(mean difference, range of differences over all the sensors, etc.). Later references to accuracy are valid (e.g. " the accuracy can be improved".. etc. is fine). AR: The LP8 sensor as integrated in the low-cost sensor unit described in this manuscript does not provide unbiased measurements in all weather conditions and is not stable over time. This means that the sensor characteristics, and the 205 performance of the outlier detection and the drift correction algorithms, and thereby the prevailing weather conditions, all have an impact on the resulting accuracy. Notably, the number of valid measurements also varies depending on the measurement filtering. Therefore, the indication of an accuracy is not straightforward. Low-cost sensors differ in this regard from highprecision instruments that usually maintain stable conditions in the measurement cell.

The error indicated in line P16L12 was based on the analysis presented by the boxplots in Figure 10. However, the reviewer is right that we should improve the respective information. Now, we indicate the 25%/50%/75% quantiles of the weekly RMSE 210 values of sensors deployed at BRM, HAE, LAEG, PAY and RIG and describe why we indicate the error by a range.

- The indication of the resulting error by a range gives a good impression of the achievable accuracy. Figure 10 clearly shows that for some weeks the accuracy is outside this range, e.g. in cases when the outlier detection algorithms do not flag all biased measurements. Figures (S6 to S9) in the supplement show the comparison of the processed sensor data and the reference
- 215 measurements. No aggregation is applied to the data sets presented in these Figures and the statistical parameters refer to the entire time period. We think that the information in the manuscript and in the supplement provides a clear picture of the achieved accuracy.

**Changes:** We rephrased the first three sentences in this paragraph beginning at P16L12.

RC: p19 L25: relation should read "relationship" Changes: We changed "relation" to "relationship".

> RC: p21 L9: This is misleading - the calibrations themselves did not account for RH dependence; rather the water vapor correction was applied, and the comparison with high-precision data showed the RH dependence was not there until a threshold was reached. Perhaps cite Fig. 3 here.

**Changes:** We rephrased the sentences of this paragraph in order to improve clarity.

**RC:** P21 L34. The authors have done such a great amount of work to evaluate these sensors, that to boil it all down to this range seems to do it disservice. Please state what this number is and how it was calculated.

**AR:** Compare our response to the RC concerning P16 L12.

**Changes:** We extended the explanations how the RMSE value was computed in section 4.2 and, based on this, we rephrased the first three sentences in the paragraph starting at P21L34. Considering the RMSE values we changed  $2\cdot\sigma$  by  $3\cdot\sigma$  in this paragraph providing now a more careful assessment of the sensors' capabilities.

**RC:** P21 L37: And where is this 20ppm shown? It is not clear where this comes from at 2-sigma - is it from the RMSE shown in the figures?

**AR:** The indicated value of 20 ppm is deduced from the RMSE value described in chapter 4.2. We reconsidered the assessment of the sensor performance and changed  $2 \cdot \sigma$  to  $3 \cdot \sigma$  providing now a more careful assessment of the sensors' capabilities. **Changes:** We changed the sentence in line P21L37.

RC: P22 L1: last sentence is awkward.

Changes: We rephrased the sentence.

**RC:** Data Availability: I encourage the authors to ensure the data is full available prior to publication. That is my understanding of the rules of this journal.

**AR:** There is a data management plan that deals with the publication of the data set. However, the release of the  $CO_2$  data including meaningful description is scheduled but still requires some more time. **Changes:** None.

RC: SI, page 11 there is a non-English word there!!Changes: We replaced the word by the English expression.

**Reviewer #3**

RC: It will be interesting to see whether all of the NDIR sensors available have similar performance characteristics as the one described here or whether there are some that have more stable in-field characteristics. If the authors have insight or speculation into that question it would be interesting to know their opinion.

**AR:** We refrain from discussing other sensor products here as this is not the scope of this manuscript and should be done together with presenting related data sets for supporting the statements.

Changes: None.

---

## Author Response (AR2)

**Reply to the editor's comment on the manuscript "Integration and calibration of NDIR CO₂ low-cost sensors, and their operation in a sensor network covering Switzerland"**

Michael Mueller[1], Peter Graf[1], Jonas Meyer[2], Anastasia Pentina[3], Dominik Brunner[1], Fernando Perez-Cruz[3], Christoph Hüglin[1], and Lukas Emmenegger[1]

[1]Empa, Swiss Federal Institute for Materials Science and Technology, Duebendorf, Switzerland
[2]Decentlab GmbH, Duebendorf, Switzerland
[3]Swiss Data Science Center, Zurich, Switzerland

*Correspondence*: Michael Mueller (michael.mueller@empa.ch)

**Editor:** Please add a couple of quick comments on (i) future pressure corrections and (ii) why RH correction did not improve noise.

**AR:** (i) We will continue the pressure correction as implemented for the processing of the measurements from the LP8 sensor units and described in this publication.

The current (second) version of the LP8 sensor unit engineered by Decentlab is now equipped with a low-cost pressure sensor. For these units, no pressure information from external modelling is required. We expect that the low-cost pressure sensors will provide accurate enough pressure information. However, this will have to be confirmed experimentally.

(ii) We added a comment in the manuscript.

**Changes:** We added a comment related to (ii) in the manuscript.